# Probabilistic hazard analysis of the gas emission of Mefite d'Ansanto, Southern Italy

Fabio Dioguardi[1,2], Giovanni Chiodini[3], Antonio Costa[3]

[1]Dipartimento di Scienze della Terra e Geoambientali, University of Bari "Aldo Moro", Bari, Italy
[2]The Lyell Centre, British Geological Survey, Edinburgh, United Kingdom
[3]Sezione di Bologna, Istituto Nazionale di Geofisica e Vulcanologia, Bologna, Italy

*Correspondence to*: Fabio Dioguardi (fabio.dioguardi@uniba.it)

**Abstract.** The emission of gas species dangerous for human health and life is a widespread source of hazard in various natural contexts. These mainly include volcanic areas but also non-volcanic geological contexts. A notable example of the latter
occurrence is the Mefite d'Ansanto area in the Southern Apennines in Italy. Here, large emissions of carbon dioxide ($CO_2$) occur at rates that make this the largest non-volcanic $CO_2$ gas emission in Italy and probably of the Earth. Given the topography of the area, in certain meteorological conditions a cold gas stream forms in the valleys surrounding the emission zone, which proved to be potentially lethal for humans and animals in the past. In this study we present a gas hazard modelling study that considers the main specie, that is $CO_2$, and the potential effect of another notable specie, which is hydrogen sulphide ($H_2S$).
For these purposes we used VIGIL, a tool that manages the workflow of gas dispersion simulations, both in the dense and dilute gas regime, specifically optimised for probabilistic hazard applications. In its latest version, VIGIL can automatically detect the most appropriate regime to be simulated based on the gas emission properties and meteorological conditions at the source. Results are discussed and presented in form of maps of $CO_2$ and $H_2S$ concentration and persistence at various exceedance probabilities considering the gas emission rates and their possible range of variation defined in previous studies.
The effect of seasonal variations is also presented and discussed.

## 1 Introduction

Carbon dioxide ($CO_2$), naturally released into the atmosphere, is a gas that can cause harm to humans and animals above certain concentration thresholds and exposure times (e.g. NIOSH, 1976; Granieri et al., 2013; Folch et al., 2017; Settimo et al., 2022 and references therein). In nature, $CO_2$ can be emitted into the atmosphere by various processes, from slow steady emissions
diffuse soil degassing or from volcanic fumaroles (e.g., Chiodini et al., 2021) to catastrophic short-lived large-volume emissions caused by limnic eruptions during lake overturns (e.g., Folch et al., 2017). Other scenarios are possible, like the emission from $CO_2$ reservoirs in certain geological contexts. This is the case of Mefite d'Ansanto, which represents the largest non-volcanic $CO_2$ gas emission of Italy and probably of the Earth (Chiodini et al., 2010). In this area in the Southern Apennines (Italy), during periods with stable atmosphere and low winds, the gas, denser than the surrounding air, is channelized at the
bottom of a W-E-trending valley, forming a lethal and invisible gas river. The frequent occurrence of the gas river is revealed

by the lack of vegetation at the bottom of the valley. Here, dead wild and domestic animals (dogs, cats, foxes, etc.) killed by the high concentration of $CO_2$ are often found. Furthermore, in the past, several lethal accidents involved humans too. Specifically, historical chronicles of XVII–XVIII century describe the death of nine people (Gambino, 1991). More recently, three people died in the 1990's (Chiodini et al., 2010).

In this work we use the computational workflow VIGIL (VolcanIc Gas dIspersion modeLling, v1.3.7, https://github.com/BritishGeologicalSurvey/VIGIL/releases/tag/v1.3.7) (Dioguardi et al., 2022) to carry out a probabilistic hazard analysis of cold $CO_2$ emissions and streams at Mefite d'Ansanto. VIGIL is a Python tool that manages the gas dispersion simulation workflow for a wide range of applications (single forecast or reanalysis simulation, multiple reanalysis simulations, probabilistic hazard assessment applications). It allows simulating both passive dispersion of a gas specie in the atmosphere

by interfacing with DISGAS v2.5.3 (Costa et al., 2009; Costa and Macedonio, 2016) (http://datasim.ov.ingv.it/models/disgas.html, hereafter referred to as DISGAS), and dense gas flows on real topography by means of TWODEE-2 v2.6 (Hankin and Britter, 1999; Folch et al., 2009, 2017) (http://datasim.ov.ingv.it/models/twodee.html, hereafter referred to as TWODEE-2). In both cases, VIGIL employs DIAGNO v1.5.0, which is a mass consistent wind model modified after DWM (Douglas et al., 1990) (http://datasim.ov.ingv.it/models/diagno.html, hereafter referred to as DIAGNO)

to simulate the meteorological conditions (wind, temperature gradients) at high resolution in the computational domain starting from observed or modelled meteorological conditions in single locations of the domain. The need to automatically managing the simulation workflow of an atmospheric dispersion application, which is a procedure that involves different and often time-consuming steps (meteorological data retrieval and processing, high-resolution meteorological simulations, gas dispersion simulation, analysis and post-processing), is particularly evident for Probabilistic Hazard Assessment (PHA) applications, in

which a usually large number of simulations are carried out in order to explore the uncertainty related to the input parameters (e.g., the wind field, the gas emission rate) (Magill and Blong, 2005; Martí et al., 2008; Neri et al., 2008; Marzocchi et al., 2010; Selva et al., 2010; Sandri et al., 2014; Mead et al., 2022). PHA carried out using gas dispersion models like TWODEE-2 and DISGAS (Costa et al., 2009; Folch et al., 2009; Costa and Macedonio, 2016) is based on multiple deterministic simulations of gas concentration in the area of interest aimed to explore not only the natural variability in input and boundary

conditions (seasonal and daily wind variability, source position and gas flux at the emission sources), but also the impact of the uncertainty on other controlling factors such as, for example, the resolution of topographic or meteorological data (Tierz et al., 2016; Selva et al., 2018; Massaro et al., 2021).

For the Mefite d'Ansanto application, Chiodini et al. (2010) performed TWODEE-2 simulations of the $CO_2$ stream only in very low wind conditions as for the period when they carried out the measurements campaign, which is when the gas river

forms. In this work, thanks to the novel features introduced in VIGIL in v1.3.7, we could explore the uncertainty related to the wide range of meteorological conditions while varying the $CO_2$ emission rates within the range of the confidence values reported in Chiodini et al. (2010). Specifically, from the latest version it is possible to automatically determine the most appropriate gas dispersion scenario (dense gas flow or dilute gas advection-diffusion) based on the gas emission properties and meteorological conditions at the source (see Section 3.1.2). In this way we could quantify the probabilistic hazard of the

$CO_2$ concentration in the Mefite d'Ansanto area, without a-priori focusing on the gas river scenario. Furthermore, knowing the chemical composition data for the Mefite gas emissions, we could also obtain a first insight on the hazard of hydrogen sulfide ($H_2S$).

    In the following we summarize the geological origin of these steady intense $CO_2$ emissions in the Mefite d'Ansanto area; then we briefly recall the features of VIGIL and its capabilities, we present the modelling strategies, the PHA outputs and their

implications for the safety of livestock and people in the area.

## 2 Geological origins of the Mefite d'Ansanto gas emissions

    Carbon dioxide dominated gas emissions and groundwaters rich in deeply derived $CO_2$ affect a large portion of the western side of the Apennine chain at the root of which the gas is thought to be generated by the melting of the carbonates of the subducting Adria plate (Chiodini et al., 2004; Frezzotti et al., 2009; Di Luccio et al., 2022). The Mefite d'Ansanto is the biggest

of the numerous cold gas emissions (>150) located in this sector of the Italian peninsula (www.magadb.net; Chiodini et al., (2010) and references therein). The isotopic signature of He and $CO_2$ ($^3$He/$^4$He ratio expressed as $R/Ra$ of 2.58 and $\delta_{13}C_{CO2}$ = +0.12‰ where $Ra$ is the helium isotopic ratio in the atmosphere, Table 1) are very similar to those of the fumaroles of the Vesuvio and Campi Flegrei volcanoes ($\delta_{13}C_{CO2}$ from −2‰ to 0.5‰, $R/Ra$ from 2.6 to 3.4; Caliro et al., 2007), a fact that suggested the presence of magmas into the axial part of the sedimentary chain (Italiano et al., 2000; Chiodini et al., 2004).

Among the gas species detected in the Mefite gas, it is worth noting the non-negligible presence of hydrogen sulfide ($H_2S$), which may be harmful or even lethal to humans at relatively low concentrations (NIOSH, 1976, 2019, 2023; OSHA, 2023) and hence can further increase the gas hazard in the area.

    In its ascending towards the surface the gas accumulates in a buried permeable structure made of limestone covered by an impermeable formation. In that zone it forms a gas pocket at a depth of about 1 km (Figure 1). This trap, reached by the Monte

Forcuso deep well at 1100-1600 m depth, contains a separated gas phase rich in $CO_2$ and feeds at the surface the Mefite gas emission (Chiodini et al., 2010 and reference therein). The conceptual scheme of Figure 1 was recently confirmed by seismic surveys (La Rocca et al. 2023).

| $CO_2$ | $H_2S$ | | $N_2$ | Ar | He | $H_2$ | $O_2$ | $CH_4$ | $\delta^{13}C_{CO2}$ (‰) | $^3$He/$^4$He |
|---|---|---|---|---|---|---|---|---|---|---|
| 980000 | 3580 | | 14300 | 10.9 | 16.7 | 80.7 | 26 | 2130 | +0.12 | 2.58 |

**Table 1. Chemical and isotopic composition of Mefite gas (concentrations in µmol/mol; Rogie et al., 2000)**


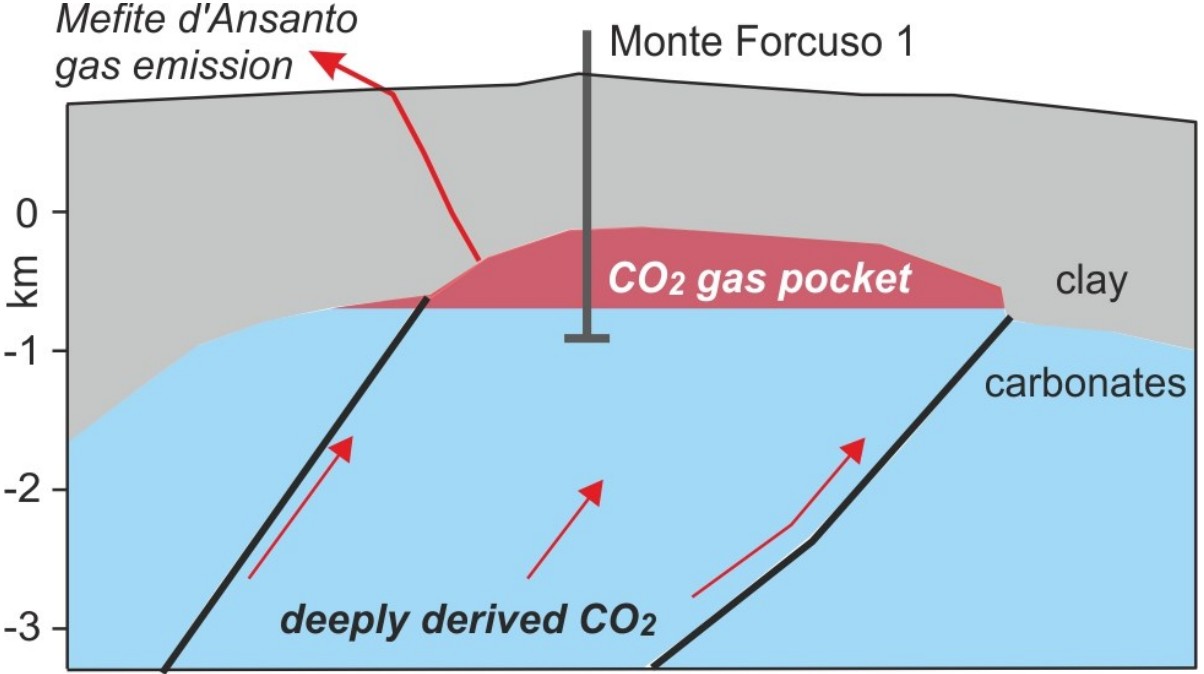

**Figure 1. Section of the system feeding Mefite gas emission (redrawn with permission from Chiodini et al. (2010). Copyright 2010 by the American Geophysical Union. Geological section from Mostardini and Merlini (1986)).**

## 3 Probabilistic hazard modelling at Mefite d'Ansanto using VIGIL

The atmospheric dispersion of a gas emitted by a natural source into the atmosphere is initially controlled by the starting density contrast between the gas and the environment (atmospheric air) and turbulent entrainment of atmospheric air driven by lateral eddies that increase the mixing with air around the edges of the plume, thereby decreasing its bulk density (Hanking and Britter, 1999; Costa et al., 2009; Folch et al., 2009; Costa and Macedonio (2016). The flow Richardson number at the source $Ri$ is a parameter that takes these factors into account:

$$Ri = \frac{1}{v^2}\left(\frac{g'q}{r}\right)^{\frac{2}{3}} \tag{1}$$

where $g'$ is the reduced gravity:

$$g' = g\frac{\rho_g - \rho_e}{\rho_e} \tag{2}$$

which quantifies the starting buoyancy of the gas phase with starting density $\rho_g$ relative to the environment with density $\rho_e$. $q$ is the source volumetric flow rate, $r$ is the gas plume radius at the source, which quantifies its extension, and $v$ is the wind speed. Based on the values of $Ri$, two regimes are possible (Cortis and Oldenburg, 2009; Costa et al., 2013):

- If Ri < 0.25, the gas transport is passive and dominated by the wind advection and diffusion (passive dispersion).

- If Ri > 1, the gas transport is dominated by the density contrast between the gas and the surrounding environment; in this case the gas flows over the topography until the density contrast persists, being the $CO_2$ at atmospheric temperature heavier than air.

Although the subsequent atmospheric gas dispersion can be theoretically simulated by solving 3D equations for the conservation of mass, momentum, and energy for each gas species, several scenario-specific simplifications are assumed in practice to reduce the computational time (i.e., single species of gas, incompressible fluid; Costa and Macedonio, 2016), eventually approaching the two end-member scenarios represented by the DISGAS and TWODEE-2 models. In both cases, the momentum coupling with the atmospheric air, which is one-way (i.e., the atmospheric wind field influences the gas cloud or stream and not vice-versa), is taken into account by considering the wind field in the computation domain (simulated with DIAGNO through VIGIL), which is the main factor controlling the light-gas dispersion and one of the controlling factors (together with density contrast) in the heavy-gas case.

DISGAS is a Eulerian model able to simulate the passive dispersal of gases in the atmosphere over complex topographic domains. It assumes that the process is governed by the wind and atmospheric turbulence and solves for the advection-diffusion equation. DISGAS allows specifying the diffusion coefficients in the three directions or using models, specifically:

- "Similarity" option for the vertical diffusion coefficient, following the Monin-Obukhov similarity theory (e.g., Monin and Yaglom, 1979; Byun 1990);
- "Smagorinsky" option for the horizontal diffusion coefficients (Smagorinsky (1963); Pielke et al. (1992); Byun and Schere (2006))

It is optionally coupled with the meteorological processor DIAGNO, which provides a mass-consistent gridded wind field from meteorological data (''observations'') in the computational domain. DISGAS supports uniform meteorological conditions too, by extrapolating time series of data from a single location in a domain (e.g., a weather station). More details on DISGAS can be found in Costa and Macedonio (2016). As we have already mentioned, in this work, in order to capture the topographic effects and the local variability, we model the wind field using DIAGNO (Douglas et al., 1990) initialized from the ERA5 Reanalysis dataset, while the meteorological variable are internally calculated by DISGAS, as explained in Costa and Macedonio (2016), by using the wind provided by DIAGNO and the ERA5 2m surface temperatures.

On the other hand, TWODEE-2 code solves a time-dependent model for the flow of a heavy gas based on the shallow layer approach. It is built on the depth-averaged equations for a gas cloud resulting from mixing a gas of density $\rho_g$ with an environment fluid (air) of density $\rho_e$ ($\rho_g > \rho_e$). TWODEE-2 is derived from the optimization and generalization of a previous Fortran-77 version developed by Hankin and Britter (1999). Under the assumption that $h/L \ll 1$ ($h$ being the gas cloud depth and $L$ a characteristic length), the 2D shallow-layer approach allows a compromise between more realistic but computationally demanding 3D CFD models and simpler 1D integral models. Such an approach is able to describe the cloud in terms of four variables: cloud depth, two depth-averaged horizontal velocities, and depth-averaged cloud density as functions of time and position. A full description of the physical model can be found in Folch et al. (2017, 2009). Like DISGAS, TWODEE-2

supports both the uniform wind and DIAGNO options (using a shallow layer approach, the field at one single reference height only is needed).

## 3.1 The gas dispersion simulation workflow with VIGIL

VIGIL (Dioguardi et al., 2022) is a Python code designed to manage the simulation workflow required to carry out numerical simulations of atmospheric gas dispersion, i.e.:

1. Retrieving and processing the meteorological data to produce the high-resolution wind field required to simulate atmospheric gas dispersion in the computational domain taking the topography into account via a Digital Elevation Model (DEM);
2. Run the simulation with the gas dispersion model;
3. Process the results to produce new outputs (e.g. probabilistic outputs) and optional plots of gas concentration.

### 3.2.1 Step 1: meteorology

The meteorological data represent the first and most important source of uncertainty variability that VIGIL explores in its current version when PHA is conducted; this implies that, if N realizations of the meteorological conditions are taken into account, VIGIL will run N dispersion simulations. Other sources of uncertainty (e.g., emission rates and gas emission source locations) can also be taken into account on top of the meteorological data variation, i.e., it is possible to apply different gas emission rates to different dispersion simulations, but these will not increase the number of the simulations N, as they are statistically sampled within the N realizations.

The source of the meteorological data depends on the chosen simulation mode (forecast or reanalysis):

- Forecast mode: NCEP (National Centers for Environmental Prediction) Global Forecast System (GFS) Numerical Weather Prediction (NWP) dataset.
- Reanalysis mode: Copernicus Climate Change Service (C3S), 2023 ERA5 NWP dataset. The user can also provide data manually, for example data from weather stations installed in the computational domain.

GFS and ERA5 are global models with meteorological data saved on a discrete grid with a typical horizontal spacing of approximately 30 km. The typical domain used in the gas dispersion applications under analysis in this work and managed by VIGIL is few km-sided, which means that the NWP data needs to be interpolated into the computational domain. Furthermore, these simulations are generally carried out at a very high spatial resolution (down to few m) in order to capture the gas clouds emitted from point sources (e.g., fumaroles) and the topographic control on gas clouds and cold gas streams. In order to produce reliable simulations of the gas dispersion in these contexts, the simple interpolation into the domain (with the assumption that the meteorological conditions are uniform and equal to the interpolated values throughout the domain) is not sufficient and a high-resolution weather prediction model is used to obtain a realistic wind field at the required spatial resolution starting from

the interpolated data. In this step VIGIL, starting from the NWP interpolated data at the centre of the domain, prepares the input data to run the mass consistent wind model DIAGNO.

### 3.1.2 Step 2: run the models

In this step, VIGIL runs DIAGNO to obtain the meteorological conditions in the computational domain required by DISGAS and/or TWODEE-2. Subsequently, it runs DISGAS or TWODEE-2 (based on the user's choice). From v1.3.7, a new option that allows the automatic detection of the scenario (light gas or heavy gas), based on the calculation $Ri$ (eq. 1) for each source in the domain, is available. This is very useful in the following situations:

- PHA applications, for which several simulations with varying meteorological conditions are carried out. In this case it is unpractical to manually a-priori determine the correct choice of the scenario and, hence, of the dispersion model (DISGAS or TWODEE-2) for each simulation.

- Single simulations or PHA applications in which multiple sources with different flow rates are present in the domain. A typical situation may be a domain containing both a weak source representing diffuse degassing, which becomes wind-dominated at very low winds, and a strong source (e.g., a fumarole), which becomes wind-dominated at higher wind speeds. In some situations, each source may behave differently as far as $Ri$ is concerned. In such a situation VIGIL v1.3.7 allows splitting the simulation in a DISGAS and a TWODEE-2 simulation, each one with their correct gas sources. It then merges the outputs of the two simulations. Furthermore, from v1.3.7 VIGIL allows varying the source emission rate using via an Empirical Cumulative Density Function (ECDF) (whose values are provided by the user in a separate input file), a uniform or a gaussian distribution. The latter feature is used in this work.

This new utility exploits a new feature introduced in DIAGNO v1.5.0 that allows tracking the wind speed at locations specified by the user. VIGIL sets the locations at the centre of the gas sources. It is important to stress that the transport regime can change as the gas flow dilutes downstream and it can no longer be treated accurately with a TWODEE-2. This change cannot be handled with the scenario-based simulation approach employed by VIGIL. Such a limitation can only be overcome by using a more complex computational fluid dynamics model, which would be more demanding in terms of computational resources and, hence, unpractical to use in PHA applications. However, as explained in the Discussion Section 5, it does not result in a significant discrepancy and, from a hazard assessment point of view, may imply a slight cautious overestimation of the hazard.

For simplicity, VIGIL v1.3.7 closes the $Ri$ gap between 0.25 and 1 by activating TWODEE-2 when $Ri > 0.25$. Future versions of VIGIL will introduce a more robust management of the automatic scenarios in this range by considering the results from the two end-members. For this study, we verified that using DISGAS or TWODEE-2 introduce a variation of the simulated gas concentrations in the domains that is not significant compared to the maximum expected values (see Section 5).

### 3.1.3 Step 3: post processing of the results

This step deals with the post processing of the DISGAS and/or TWODEE-2 results. Various functionalities are available:

- Conversion between the tracked gas specie into other gas species provided gas species properties (e.g., molar weights, molar ratios between the converted and the tracked species) are available in a separate file.
- Production of time series of gas concentration at selected locations (tracking points).
- For PHA applications, generation of ECDFs of the gas concentration and extrapolation of the gas concentration at the user's desired exceedance probability. The ECDF is calculated by merging, at each point of the domain and for each time step, the concentrations obtained in all simulations using the Python NumPy quantile function.
- For PHA applications, if gas concentration thresholds and exposure times are provided in input, persistence probability (i.e., the probability to overcome a certain threshold for its exposure time) is calculated.
- For PHA applications, if the tracking points functionality is activated, hazard curves (i.e., exceedance probability vs. gas concentrations) are calculated.

Plots of all these outputs can be optionally generated.

## 4. Numerical simulations of gas flow at Mefite d'Ansanto and probabilistic outputs

### 4.1. The PHA workflow

In this section we review the workflow followed to carry out the PHA at Mefite d'Ansanto. All the input files and VIGIL commands required to reproduce the workflow and the outputs are available in the online Zenodo repository (Dioguardi, 2023). Details on the input files and commands can be found in the VIGIL User manual and Dioguardi et al. (2022).

We carried out a PHA at Mefite d'Ansanto by exploring the meteorological data variability as the main source of uncertainty and, for each simulation, automatically setting the gas emission rate sampling it from a normal distribution with mean 23.1 kg $s^{-1}$ and standard deviation 5.8 kg $s^{-1}$ according to Chiodini et al. (2010). $CO_2$ is emitted from an area that can be roughly approximated by a square of 3500 $m^2$, which would correspond to a radius $r$ of 33.4 m of the equivalent circle (Figure 2). Assuming thermal equilibrium between the emitted gas and the atmosphere, at 15 °C the two species ($CO_2$ and air) have a density $\rho$ of 1.87 and 1.22 kg $m^{-3}$, respectively. The mean volumetric flow rate $q$ of the gas, based on the mean mass flow rate, is therefore 12.35 $m^3$ $s^{-1}$.

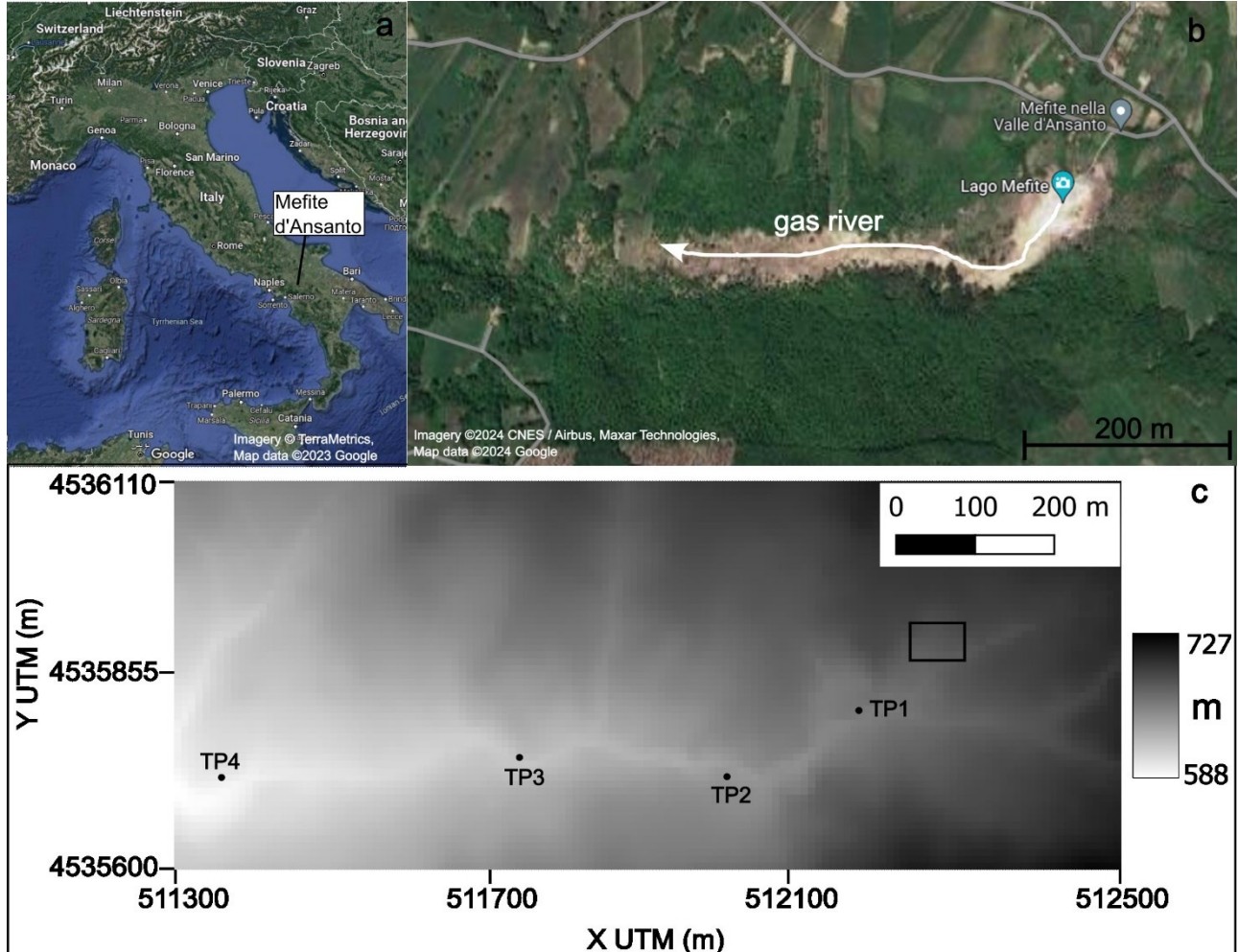

Figure 2: a) Location of Mefite d'Ansanto in Southern Italy. Imagery © TerraMetrics, Map data ©2023 Google. b) Aerial photo showing the gas emission area and the valley where the gas river forms, which is emphasized by the lack of vegetation. Imagery ©2024 CNES / Airbus, Maxar Technologies. Map data ©2024 Google. c) Computational domain with the elevation (m a.s.l.) shown in greyscale. The $CO_2$ emission area as approximated in the simulations is represented by the black rectangle. Black dots represent the locations of the four tracking points (TP) where the hazard curves are extrapolated from the simulation outputs; the coordinates of the tracking points are listed in Table 4.

Meteorological conditions were retrieved from the ERA5 dataset by using the weather.py script of VIGIL. Specifically, since we run one day-long simulations with a time step of 1 hour, we sampled 1,000 days from the period 01/01/1993 – 01/01/2023 and for each day we downloaded pressure level (Hersbach et al., 2018a) and surface data (Hersbach et al., 2018b) for a location positioned towards the centre of the computational domain. With these data, VIGIL created the input files necessary to run DIAGNO for each day.

Subsequently, we executed `run_models.py` of VIGIL to first obtain the simulated meteorological wind field over 24 hours with a time step of 1 hour for the 1,000 days in the computational domain (Figure 2) using DIAGNO. The domain extended from 511300 to 512500 m in the $X$ direction and from 4535600 to 4536110 in the $Y$ direction (UTM coordinates, WGS 84 /

UTM zone 33N) and 500 m along the vertical and was discretized with 3 m-sized square cells along the horizontal axes and with a variable vertical spacing along the vertical direction: 1, 2, 5, 10, 20, 50. 100, 250, 500 m. Then we selected the option to automatically detect which scenario (heavy or light gas) was more appropriate for the simulated day based on the gas emission characteristics and the wind conditions at the source location at 10 m above the ground. It is to note that the emission rate varied across the 1,000 days, sampling it from the normal distribution introduced above. In fact, with these gas source emission characteristics described above, the gas source satisfies the conditions of the heavy gas regime in case of very low to no-wind conditions. Specifically, based on eq. 1, the maximum wind $v$ that would satisfy the heavy gas regime ($Ri > 1$) would be 1.25 m s$^{-1}$. With a wind of 2.49 m s$^{-1}$ the passive wind-dominated conditions ($Ri < 0.25$) would be already satisfied. It resulted that 313 over 1000 runs could be simulated with TWODEE-2 (heavy gas scenario) and the rest (687 simulations) with DISGAS.

Finally, we run `post_process.py` to produce the requested outputs. Specifically, we requested:

- ECDFs' 50%, 16% and 5% exceedance probability of the $CO_2$ concentration at 2 m above the ground for the time-averaged solution over the whole simulation duration (24 hours). Furthermore, by means of the gas specie conversion capability of VIGIL, we calculated the same outputs for $H_2S$ by using the chemical composition data listed in Table 1, specifically the molar ratio between $H_2S$ and $CO_2$ (0.0036). However, we need to stress that such a conversion neglects possible sources of sinks of the converted species in the domain. Specifically, the reaction with OH radicals represents the main sink of $H_2S$ in the atmosphere (e.g., Watts, 2000), together with other minor sinks that act on a local scale, e.g., during and subsequent to rainfall events (Kristmannsdottir et al., 2000, Thorsteinsson et al., 2013) or under the action of lakes, soils, and vegetation (Bussotti et al., 1997; Cihacek and Bremner, 1990). However, these interactions typically do not play a major role as demonstrated, for example, by Olafsdottir et al. (2014), who carried out ad hoc measurement campaigns in Iceland showing that the depletion of $H_2S$ from the atmosphere is insignificant compared to the emissions within a 35 km distance from the sources. Neglecting such reactions could imply an overestimation of the $H_2S$ concentration, but we do not expect a significant effect in our restricted domain.

  It is worth noting that the current version of VIGIL does not allow starting the simulation from a pre-existent solution of the gas concentration field, i.e., each simulation starts with clean air (apart from background concentrations that the user may specify). This may affect the simulations' outputs in the initial time steps in scenarios like those under analysis in this work, i.e., with a steady long-lived gas emission. However, in a small domain like the one under consideration, the $CO_2$ cloud or stream can be considered fully developed within the first hour, which is the minimum time resolution for outputs currently allowed by VIGIL due to the limitations of DIAGNO. Therefore, we can safely assume that the effect of starting from clear air is negligible in this application when we calculate the 24-hour time-average of the outputs.

- Persistence outputs, the probability to overcome the concentration thresholds for their respective times as specified in the input file specifying the gas property (gas_properties.csv). The concentration thresholds and exposure times for $CO_2$ used in this work are listed in Table 2 and are compiled based on NIOSH (1976, 2019, 2020), Costa et al.

(2008), Granieri et al. (2013), Settimo et al. (2022) and references therein. For 1000 and 3500 ppm there are no time exposure indicated, therefore we calculate the persistence for 24 hours. For 5000 ppm, TWA (Time Weighted Average) values for $CO_2$ are commonly used in occupational health and safety to establish permissible exposure limits or recommended exposure limits. These limits define the maximum allowable concentration of $CO_2$ that a worker or individual can be exposed to over a specific time period, usually 8 hours or 24 hours. In this work we take 8 hours as exposure time. For the highest concentrations taken into account (15000, 30000 and 100000), the exposure times that cause harm are always in the range of 10-15 minutes, with the effects on humans listed in the table. In this study, for computational limitations, as reported in Table 2, the minimum exposure time we considered is 1 hour. For $H_2S$ we used the thresholds and exposure times defined by the United States Occupational Safety and Health Administration (OSHA) (https://www.osha.gov/hydrogen-sulfide/), based on the recommendations of The National Institute for Occupational Safety and Health (NIOSH, 1976, 2019, 2020) and the base threshold defined in Olafsdottir and Gardarsson, (2013). OSHA defines three limits: Recommended Exposure Limit (REL), Permissible Exposure Limit (PEL) and Immediately Dangerous to Life and Health (IDLH) exposure limit.

| $CO_2$ concentration threshold (ppm) | Exposure time in the literature | Effects | Tested exposure time in this work (hours) |
|---|---|---|---|
| 1000 | no time indicated | no effects under this threshold (Settimo et al., 2022) | 24 |
| 3500 | no time indicated | no effects if ventilated ambient (Granieri et al., 2013) | 24 |
| 5000 | TWA 8 hours | Above TWA: Slight increase in breathing rate (Costa et al., 2008; Granieri et al., 2013). | 8 |
| 15000 | Above 10 minutes | Breathing deeper and more frequent (Costa et al., 2008; | 1 |

| | | Granieri et al., 2013) | |
|---|---|---|---|
| 30000 | Above 15 minutes | Breathing increases to twice normal rate, weak narcotic effect, headache for long time exposure (Costa et al., 2008; Granieri et al., 2013) | 1 |
| 100000 | 10-15 minutes | Respiratory distress with loss of consciousness and death in 10–15 min (Costa et al., 2008; Granieri et al., 2013) | 1 |

**Table 2. CO₂ concentration thresholds and exposure times that can cause harm to humans**

| $H_2S$ concentration threshold (ppm) | Exposure time in the literature | Effects | Tested exposure time in this work (hours) |
|---|---|---|---|
| 0.035 | no time indicated | No effects reported under this threshold | 24 |
| 10 | Construction 8-hour Limit; Shipyard 8-hour limit: 10 ppm (OSHA PEL) | No effects reported under this threshold | 8 |
| 50 | General Industry Peak Limit (OSHA PEL), 10 minutes. | At 50-100 ppm, possible effects include slight conjunctivitis ("gas eye") and respiratory tract irritation after 1 | 1 |

| | | hour. May cause digestive upset and loss of appetite. | |
|---|---|---|---|
| 100 | NIOSH IDLH. Coughing, eye irritation, loss of smell after 2-15 minutes (olfactory fatigue). | Altered breathing, drowsiness after 15-30 minutes. Throat irritation after 1 hour. Gradual increase in severity of symptoms over several hours. Death may occur after 48 hours (OSHA). | 1 |
| 500 | Staggering, collapse in 5 minutes. | Serious damage to the eyes in 30 minutes. Death after 30-60 minutes. (OSHA) | 1 |

**Table 3. $H_2S$ concentration thresholds and exposure times that can cause harm to humans**

- Hazard curves, i.e., exceedance probability vs. $CO_2$ concentration plots, at four selected locations or tracking points (TP, see Figure 2) spread along the valley. Table 4 lists the coordinates of the four tracking points:

| TP# | X UTM (m) | Y UTM (m) | Z (m above the ground) |
|---|---|---|---|
| 1 | 512181 | 4535810 | 2.0 |
| 2 | 512022 | 4535724 | 2.0 |
| 3 | 511772 | 4535749 | 2.0 |
| 4 | 511413 | 4535723 | 2.0 |

**Table 4. Coordinates (UTM) and elevations (m above the ground) of the four tracking points used for the hazard curves**

We also carried out a seasonal analysis, i.e., we produced the aforementioned outputs for each season (winter, spring, autumn, summer) to check if there is a control of the season on the $CO_2$ dispersion pattern.

Figure 3 shows the concentration of $CO_2$ in the domain at 2 m above the ground at 50%, 16% and 5% exceedance probability. The concentration is obtained by interrogating the ECDF of the 24-hour time-averaged solution in each point of the domain.

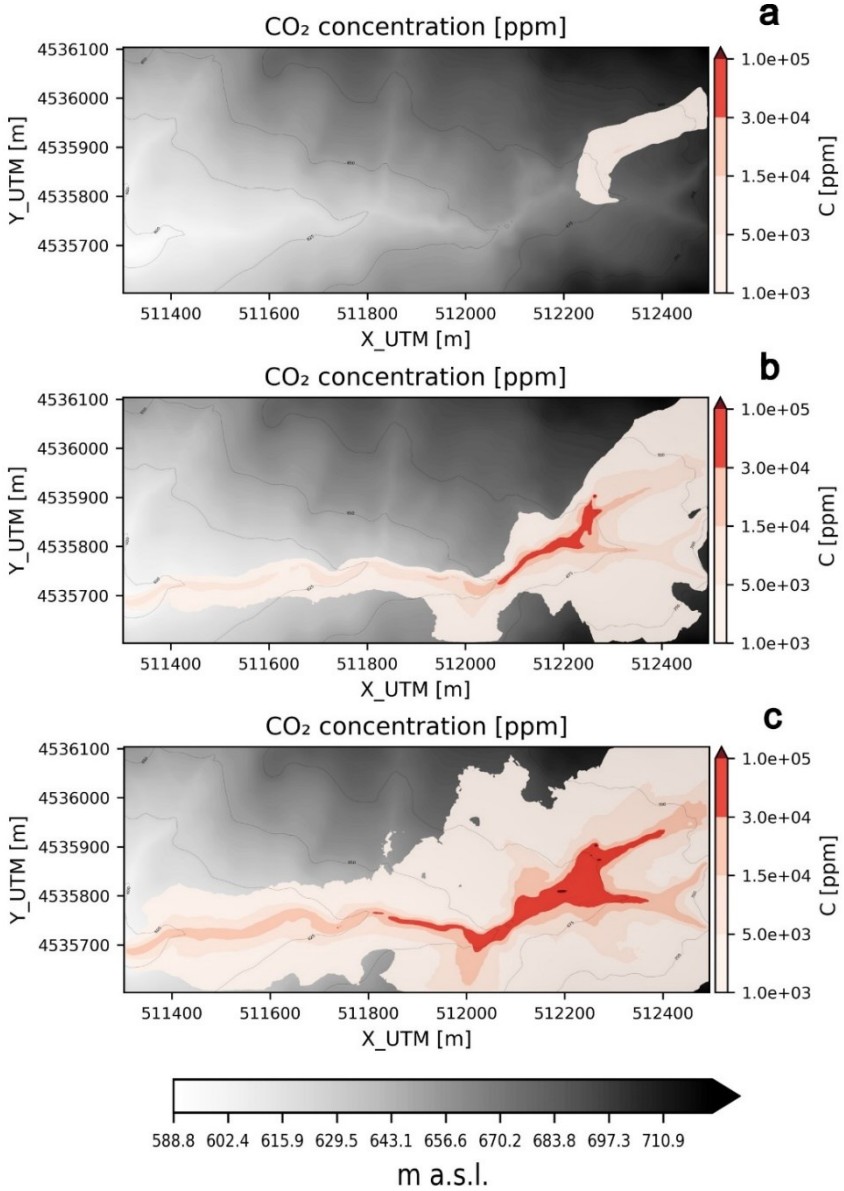

**Figure 3. 24 hour time-averaged $CO_2$ concentration at 2 m above the ground at 50% (a), 16% (b) and 5% (c) exceedance probabilities**

A likely situation, corresponding at 50% exceedance probability (Figure 3a), that is the median of the ECDF, shows non-negligible to high (few thousands of ppm) concentrations of $CO_2$ in the emission area and towards the eastern part of the

domain. These areas are uphill the emission area, which means that on average elevated $CO_2$ levels caused by the action of winds blowing from W-SW cannot be ruled out. At 16% exceedance probability (Figure 3b), which corresponds to the median + 1 standard deviations of the ECDF, the $CO_2$ concentration significantly increases up to dangerous levels (> 15,000 ppm) and

the streams along the valleys become visible. By further decreasing the exceedance probability to 5%, which corresponds to the median + 2 standard deviations of the ECDF (Figures 3c), the extent of the areas affected by $CO_2$ concentration > 15,000 ppm further increase and eventually the areas become not confined to the valleys. Areas affected by $CO_2$ concentration > 30,000 ppm extend significantly outside the emission area at 5% exceedance probability.

Figure 4 displays the persistence maps for the six concentration thresholds and exposure times listed in Table 2. All

probabilities are calculated at 2 m above the ground. Figure 4a shows the probability to overcome a $CO_2$ concentration of 1,000 ppm for at least 24 hours, which is the duration of the simulation. This probability is high in the emission area but is also not negligible (5 - 10%) along the main E-W valley and some areas towards NE in the domain. This is in line with what is observed in Figure 3a, which was interpreted as the formation of $CO_2$ plumes drifting NE due to the action of winds blowing from W-SW. Figure 4b shows the probability to overcome a $CO_2$ concentration of 3,500 ppm for at least 24 hours. As expected, the

probability is lower than the case shown in Figure 3a but still not negligible along the main valley and in the emission area and surroundings. Considering an exposure time of 8 hours, the probability to overcome a $CO_2$ concentration of 5,000 ppm is high in the emission area and surroundings (up to 50%), lower but still notable towards E-NE (up to 30%) and not negligible along most of the valleys (10 - 20%) (Figure 4c). Focusing on high (15,000 – 30,000 ppm) to very high (100,000) concentrations for an exposure time of 1 hour (Figure 4d, 4e and 4f, respectively), the extent of the domain affected by significant probabilities

is noteworthy for 15,000 ppm and 30,000, with probabilities up to 50% - 60% in the emission area and surroundings (particularly towards NE) and along the valleys (up to 20 – 30%). Even in the most extreme case of $CO_2$ concentrations > 100,000 ppm there is a 30% probability to overcome this concentration for 1 hour in the emission area and surroundings and a not negligible (up to 10 – 20 %) one along the valleys.

The hazard curves of the 24 hours-averaged $CO_2$ concentration produced at the four locations identified by the four tracking

points listed in Table 4 and shown in Figure 2 are represented in Figure 5. In the location TP1, which is the closest one to the emission area, higher concentrations of $CO_2$ than the other locations are expected, i.e., for a fixed exceedance probability the concentration is significantly higher. The maximum concentration is close to 140,000 ppm. The curves at the other locations show similar values, though slightly higher for TP2, which is the second closest to the emission area. By looking at Figure 2c, TP2, TP3 and TP4 are positioned along the main W-E valley. The fact that the concentrations are almost the same, especially

in TP3 and TP4, can be interpreted as the $CO_2$ stream maintaining its characteristics along the valley, at least in the analysed domain.

We wanted to analyse also the seasonal control on the simulations output, i.e., if the predicted concentrations differ significantly among the four seasons. We first split the dataset of simulations in four categories based on the month of each simulated day: winter (December, January and February), spring (March, April, May), summer (June, July, August) and

autumn (September, October, November). For each season, we computed the same outputs (ECDFs and persistence) as those

shown in Figures 3 and 4. In particular, Figure 6 shows the 24 hours' time-averaged $CO_2$ concentration at an exceedance probability of 16% for the four seasons. The concentrations are significantly higher in the summer season, followed by autumn. Winter and spring are characterized by similarly lower concentrations, although the concentrations are slightly lower in winter than in spring. A closer look at the plots also shows how the summer is characterized by significantly higher values of $CO_2$ concentration in the main W-E valley (Figure 6c), followed by autumn (Figure 6d). In order to better understand this seasonal control, which in our hypothesis depends on the wind speeds, we calculated the 24 hours' time-averaged domain-averaged wind speed at 10 m above the ground for each day and then, by collecting these data for each season, we calculated the ECDF of this averaged wind speed. Figure 7 shows the 24 hours' time-averaged average wind speed vs. the exceedance probability. Upon a first look at the four curves, it is evident how the wind speed is significantly lower in the summer (grey solid line), followed by the autumn (yellow solid line). The other two seasons display similar trends, although spring (orange solid line) is characterized by lower winds, particularly at exceedance probabilities lower than 50%. Summarizing the findings shown in Figure 7, it is more likely to experience higher wind intensities and hence lower concentrations in winter, followed by spring and autumn. Summer shows the lowest winds and the highest concentrations.

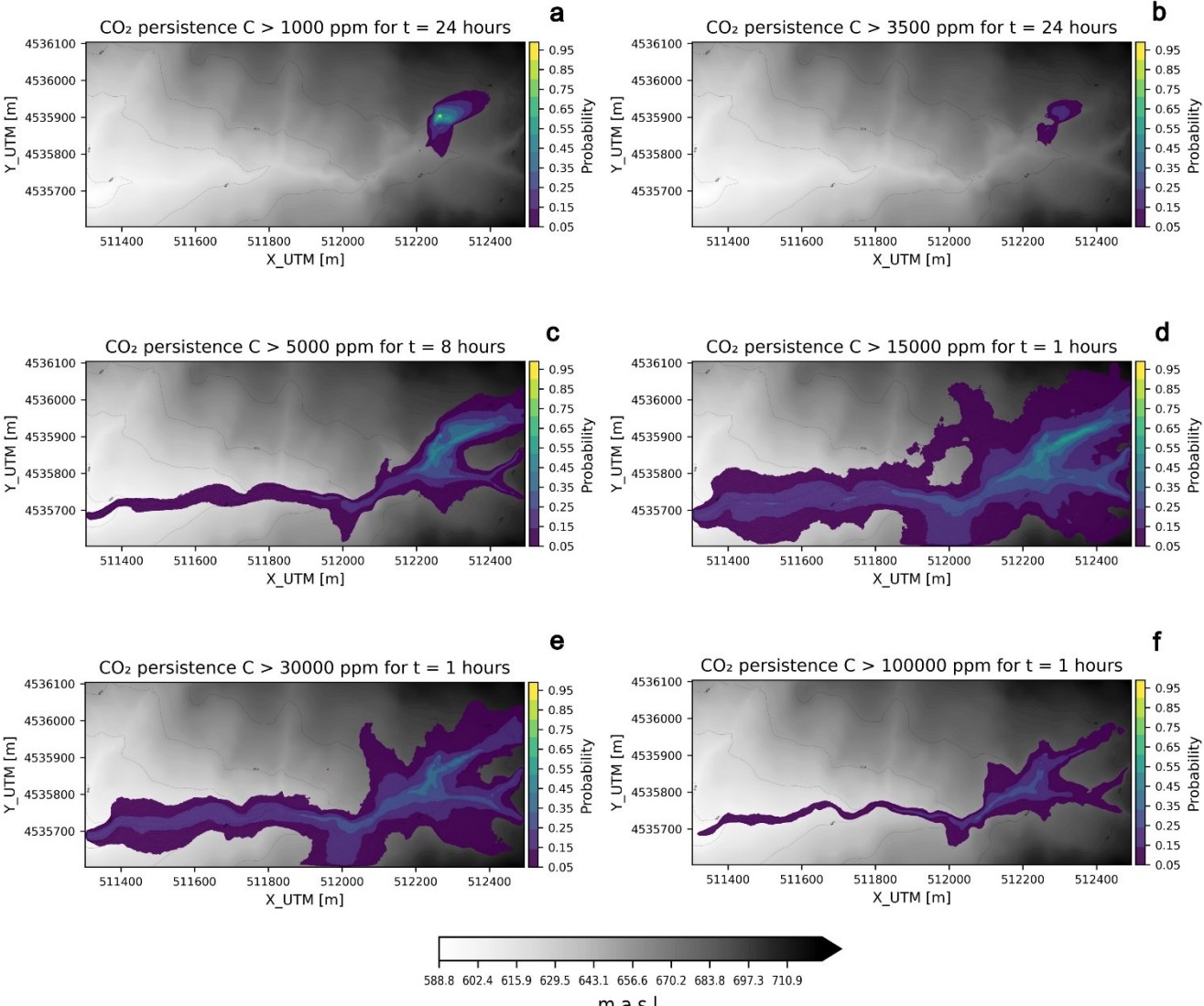

Figure 4. Persistence maps at 2 m above the ground. a) Concentration threshold = 1000 ppm, exposure time = 24 hours. b) Concentration threshold = 3500 ppm, exposure time = 24 hours. c) Concentration threshold = 5000 ppm, exposure time = 8 hours. d) Concentration threshold = 15000 ppm, exposure time = 1 hour. e) Concentration threshold = 30000 ppm, exposure time = 1 hour. f) Concentration threshold = 100000 ppm, exposure time = 1 hour.

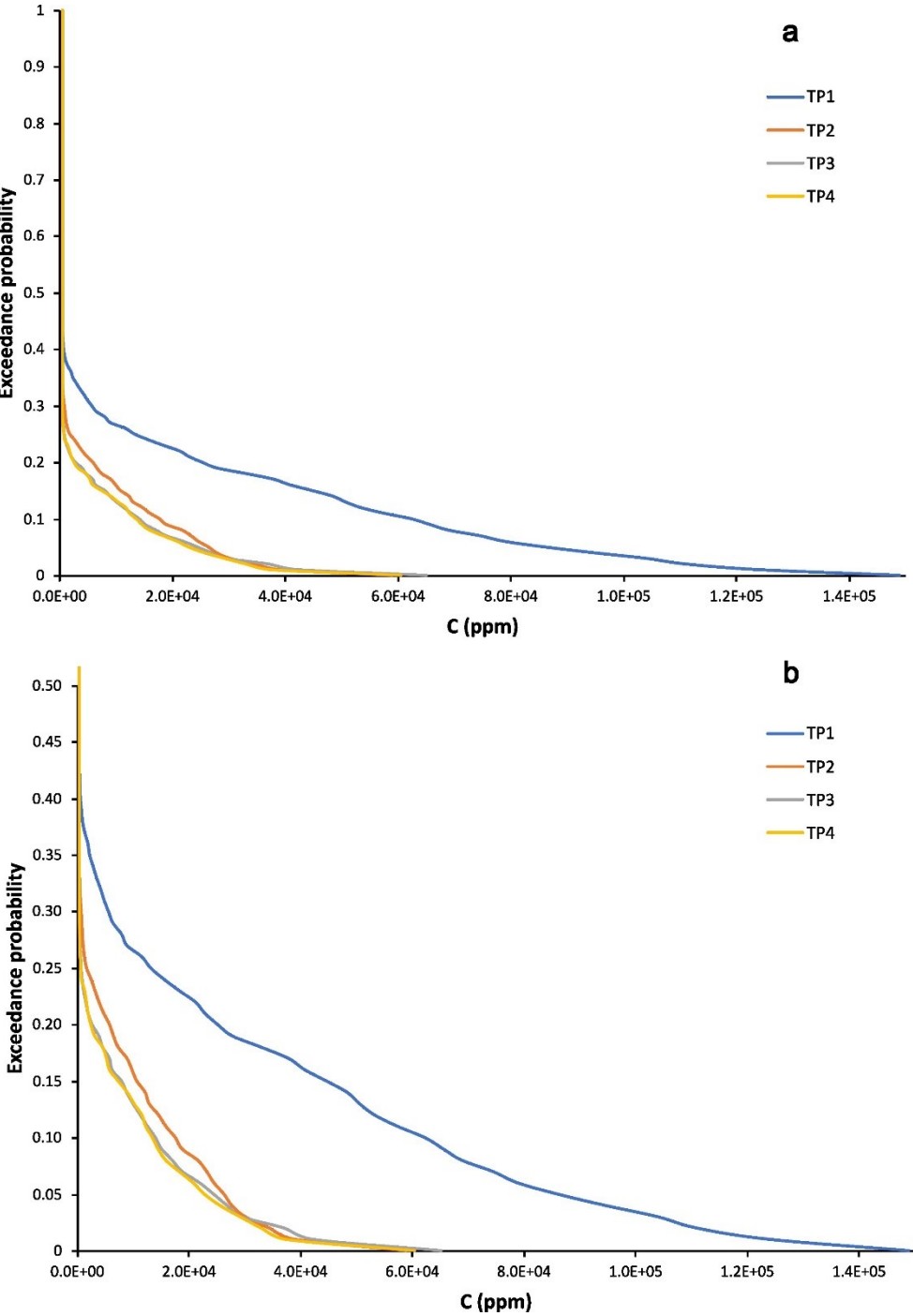

**Figure 5. Hazard curves of the 24 hours-averaged concentration at the four locations identified by the four tracking points. The hazard curves for the tracking points 1, 2, 3 and 4 are represented by the blue, orange, grey and yellow solid line, respectively. a) Plot in the original scale used by VIGIL. b) Zoomed version.**

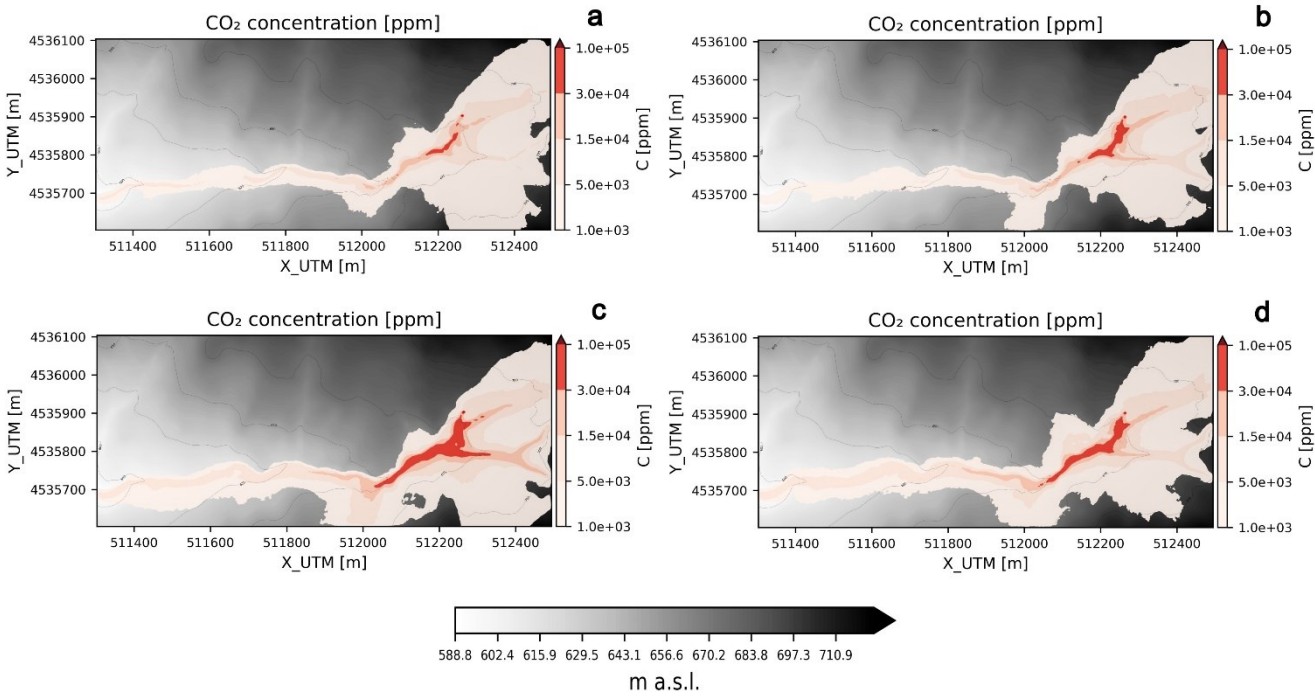

**Figure 6. 24 hours' time-averaged CO₂ concentration at 2 m above the ground at 16% exceedance probability for the four seasons:**
**a) winter; b) spring; c) summer; d) autumn.**

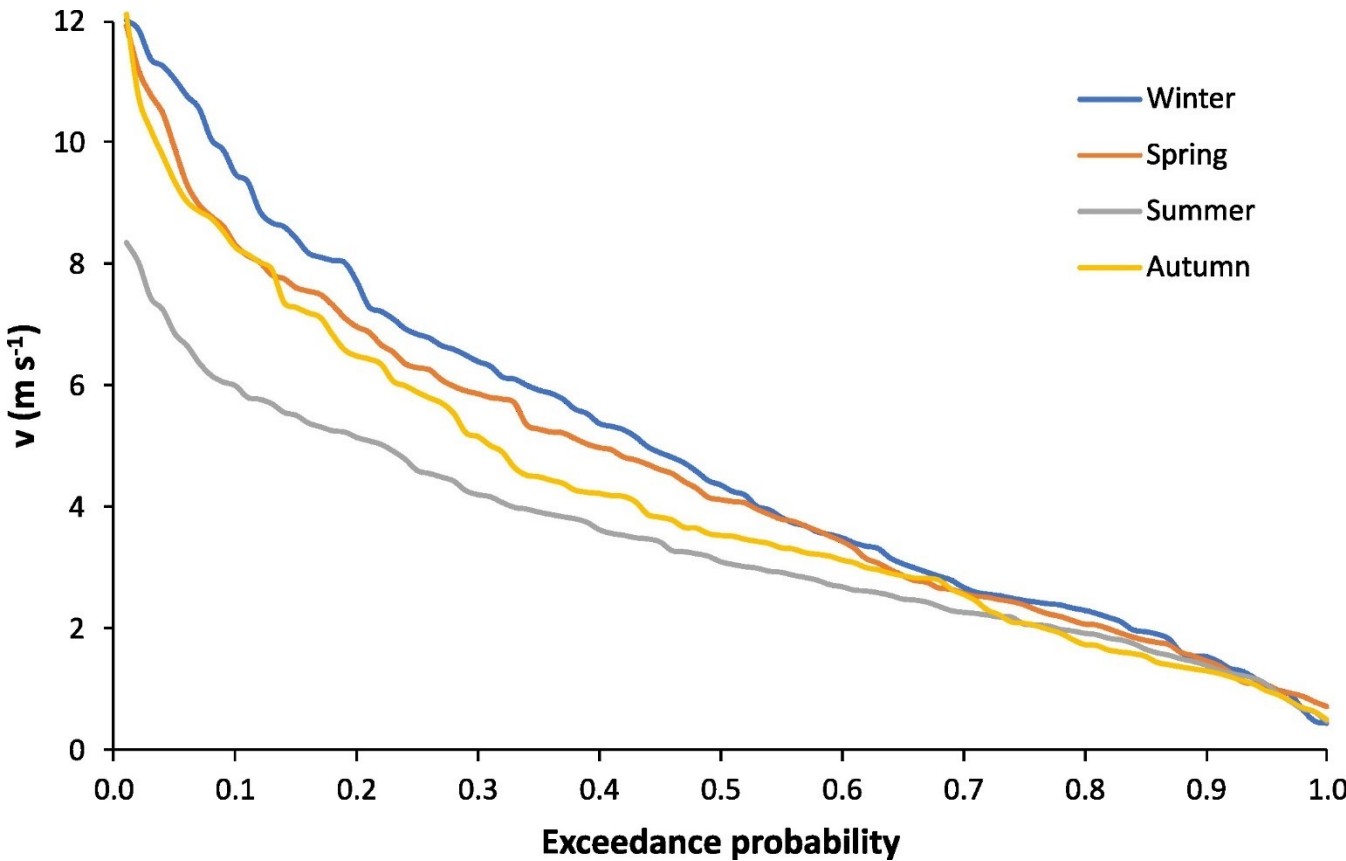

**Figure 7. ECDF of the 24 hours' time-averaged domain-averaged wind speed at 10 m above the ground for winter (solid blue line), spring (solid orange line), summer (solid grey line) and autumn (solid yellow line).**

It is also worth noting that the probabilities for winds lower than 3 m s$^{-1}$ are very similar, as for the highest winds (about 12 m s$^{-1}$), although in the latter case the summer never reaches winds higher than 9 m s$^{-1}$. The results of the analysis of the meteorological conditions explain the seasonal differences shown in Figure 6. In the summer, when high winds are less likely than the other seasons, it is more likely to form a dense gas flow river in the valleys and to have higher concentrations in the areas surrounding the gas source. Autumn shows a similar pattern, although with a slightly lower probability to produce a

dense gas flow in the valleys. This likelihood becomes significantly lower in the autumn and spring. Additionally, the results we are discussing are 24 time-averaged. In our view this is one of the reasons why the effect of winds, which is evident in the seasonal control, prevail over other effects that may dominate in specific time windows of a day (e.g., the effect of temperature inversion, which can occur under stable conditions especially in winter and enhances the gas concentration at the lowest levels, may play a role during the night and early morning).

Finally, Figure 8 shows some of the probabilistic outputs produced for the H$_2$S gas specie and the concentration thresholds and exposure times listed in Table 3. The map of H$_2$S concentration at 2 m above the ground at an exceedance probability of 5%

shows levels of $H_2S$ above one of the PEL (>50 ppm) along all the valleys and the emission area. Dangerous levels (>100 ppm) are also significantly widespread in the emission area and the adjacent sectors of the valleys. Figure 8b and c displays the persistence maps for two PELs (10 ppm for 8 hours and 50 ppm for 1 hour, see Table 3). The probability to overcome these

levels for the defined time intervals are significant in the same areas, reaching up to 50-60% in the vicinity of the emission zone for both PELs. The probability to overcome 50 ppm for 1 hour is up to 30% even 800-1000 m away from the gas source in the main W-E valley. However, we need to highlight that $H_2S$ in the atmosphere tends to react with OH radicals (e.g., Watts, 2000), and other minor sinks can be found on a local scale (e.g. rainfall events, presence of lakes, soils, and vegetation; Kristmannsdottir et al., 2000, Thorsteinsson et al., 2013, Bussotti et al., 1997; Cihacek and Bremner, 1990). Neglecting such

reactions could result in an overestimation of the $H_2S$ concentration, probably not significant for our restricted domain. Therefore, whilst further $H_2S$-specific measurement campaigns are required in order to draw conclusions on the $H_2S$ hazard, these results show that the $H_2S$ hazard should also be taken into account on top of the $CO_2$ one.

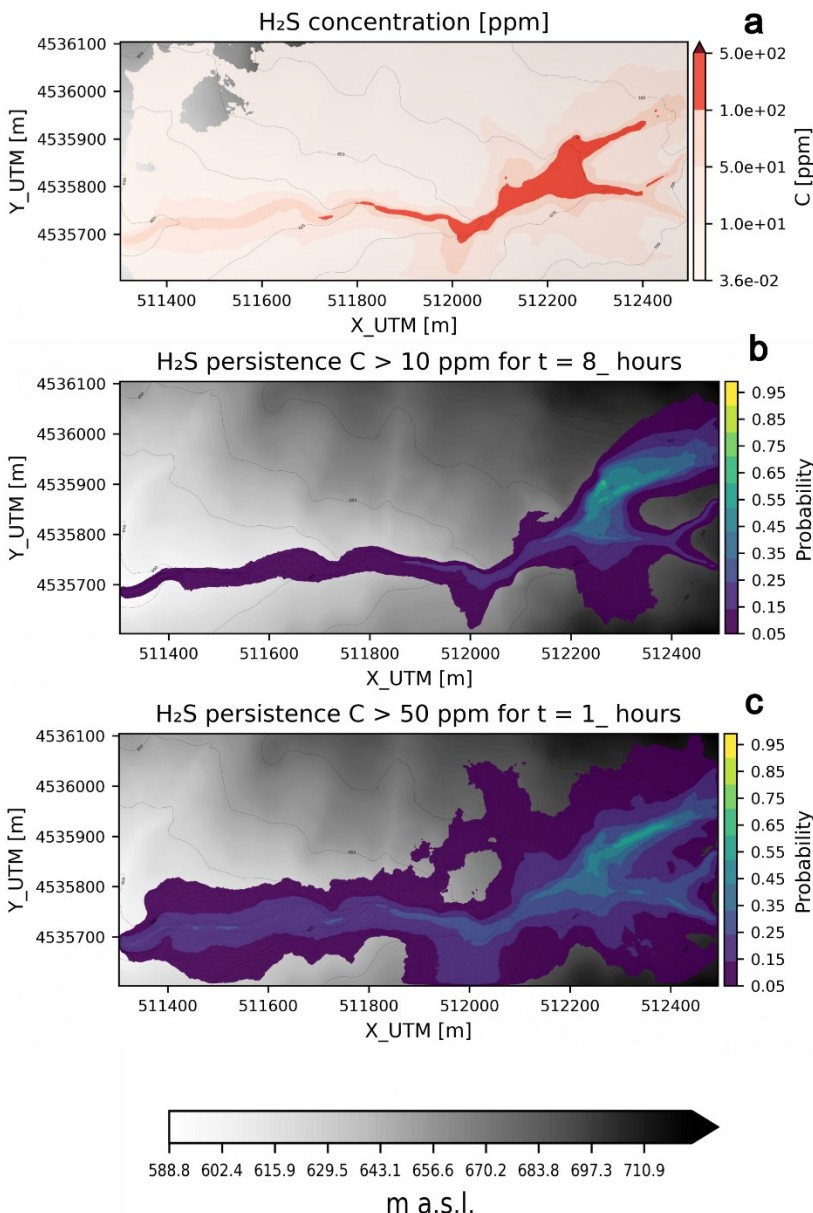

**Figure 8. H₂S probabilistic hazard maps at 2 m above the ground. a) 24 hours' time-averaged concentration at 5% exceedance probability. b) Persistence maps for PEL 10 ppm, exposure time = 8 hours. c) Persistence maps for PEL 50 ppm, exposure time = 1 hour.**


## 5. Discussion

The analysis carried out with VIGIL confirmed that the computational domain under analysis is prone to non-negligible likelihoods to be exposed to high concentrations of $CO_2$. This is particularly evident in the areas surrounding the gas source and the valleys, especially the main W-E valley where the generation of a cold $CO_2$ gas river is a well-known occurrence and marked by the absence of vegetation at the lowermost levels.

By looking at the 50th percentile of the solutions (50% exceedance probability of the ECDF generated by VIGIL, Figure 3), one can observe that on average the source area and the area towards NE from the emission zone is the most affected one, with 24 hours' time-averaged $CO_2$ concentrations up to few thousand ppm. The generation of the cold gas river in the valleys of the domain is observed at lower exceedance probabilities, which implies that this occurrence is less likely. However, the gas river is already well visible at 16% exceedance probability with very high $CO_2$ concentrations (>15,000 ppm) in many areas of the valleys and source area and surroundings. This means that there is the 16% probability to have even worse scenarios, which is not a low likelihood. In fact, at lower exceedance probabilities (5%) the scenarios look significantly worse. The persistence maps (Figure 4), created by VIGIL based on the concentration thresholds and related exposure times of Table 2, corroborates these findings. Likelihoods to overcome dangerous $CO_2$ concentrations levels for specified times are non-negligible to significant. For example, for concentrations of 30,000 and 100,000 ppm, which are dangerous to very dangerous for the humans' health and life even for exposure times of few minutes, the probability to overcome these thresholds for 1 hour are significant (up to 40-50 %) especially in the emission area and the valleys' segment close to the gas source. It is worth noting that the exposure times for these two concentrations thresholds are set to 10-15 minutes (Table 2), but we could calculate the persistence for time intervals of at least 1 hour the current computational limitations, which imposes the minimum time step to 1 hour. This means that the probabilities shown in Figure 4 (d, e, f) would likely be even higher if the proper exposure times (10-15 minutes, Table 2) would have been taken into account. Therefore, our persistence calculations results represent a lower estimate than the real one.

Similar considerations can be done for $H_2S$ (Figure 8), whose concentrations in our study were estimated by means of the gas composition data (Table 1) and the specie conversion capability of VIGIL. Results shown in Figure 8 ($H_2S$ concentration at an exceedance probability of 5% and persistence maps based on the PEL defined in Table 3) demonstrated that the hazard posed by $H_2S$ cannot be discarded in the area and should be taken into account on top of the $CO_2$ for assessing the gas hazard in this area.

Furthermore, it is worth reminding that for this study we made use of the new capability introduced in VIGIL v1.3.7, which automatically assign the gas dispersion scenario based on the $Ri$ value at the source. In the currently implemented approach, a dense-gas scenario is assigned to all the cases in which $Ri > 0,25$ in the intermediate regime ($0.25 < Ri < 1$), hence it simulated all these cases with TWODEE-2. This is a cautious choice for the hazard quantification, since TWODEE-2 generally results in higher concentrations. The number of simulations in this intermediate regime is 277, which corresponds to 27.7% of all the simulations; therefore, we conducted an analysis of the outputs produced by the two models in this intermediate regime.

Specifically, we selected three simulations with the following values of $Ri$, chosen to be equally spaced between 0.25 and 1: 0.438, 0.625 and 0.812. For each case, we conducted the simulation using both DISGAS and TWODEE-2. From the outputs, we calculated the Root Mean Square Error ($RMSE$) between the DISGAS and the TWODEE-2 solutions in the domain for all the vertical levels and time steps, then we calculated the average of all the RMSEs for each tested case:

$$RMSE = \sqrt{\frac{\Sigma_{i=1}^{N}(C_{i,DISGAS}-C_{i,TWODEE-2})^2}{N}} \qquad (3)$$

We then compared $RMSE$ with the Mean Value ($MV$) of the concentration in the domain. Since the $RMSE$ was calculated by comparing the outputs of the two models for each $Ri$ scenario, $MV$ was calculated by:

$$MV = \frac{MV_{DISGAS}+MV_{TWODEE-2}}{2} =$$

$$\frac{\frac{\Sigma_{i=1}^{N}C_{i,DISGAS}}{N}-C_{background}+\frac{\Sigma_{i=1}^{N}C_{i,TWODEE-2}}{N}-C_{background}}{2} \qquad (4)$$

Where $C_{background}$ is the $CO_2$ background concentration in the atmosphere (400 ppm). Results are summarized in Table 5 and show that the error is not significant, since it is always less than 5% of the $MV$. As expected, the discrepancy constantly increase as $Ri$ increases from 55 to 78 ppm. To be further conservative, we considered only the part of the domain where at least one of the two models computed concentration values above the background concentration used in the simulations (400 ppm). In this case we obtained larger RMSE values, but still not significant compared to the $MV$. These results corroborate our modelling approach in the intermediate regime for this specific case of very high concentrations. Future versions of VIGIL will introduce more robust approaches.

| $Ri$ | $RMSE$ All domain [ppm] | $MV$ All domain | $RMSE / MV$ [%] | $RMSE$ Reduced domain [ppm] | $MV$ Reduced domain | $RMSE / MV$ [%] |
|---|---|---|---|---|---|---|
| 0.438 | 54.71 | 1246.25 | 4.39 | 102.38 | 4360.19 | 2.35 |
| 0.625 | 60.84 | 1223.66 | 4.97 | 159.56 | 8051.21 | 1.98 |
| 0.812 | 77.82 | 2326.30 | 3.35 | 155.10 | 9576.85 | 1.62 |

Table 5. Results of the analysis of the influence of selecting either DISGAS or TWODEE-2 in the intermediate $Ri$ regime.

Finally, since DISGAS and TWODEE-2 outputs depend on the turbulent diffusion, which in turn depends on the spatial resolution of the computational domain, we carried out a test to verify the dependency of the solution on the chosen spatial

discretization of the domain. Results show that, with our settings, both DISGAS and TWODEE-2 outputs are not significantly affected by the chosen resolution (see Appendix A).

## 6. Conclusions

In this work we presented results of the first PHA carried out at Mefite d'Ansanto, the largest non-volcanic $CO_2$ gas emission of Italy and probably of the Earth. To do so, we used VIGIL v1.3.7, a Python tool designed to manage the workflow of gas dispersion simulations and post-processing, specifically designed to carry out PHA applications. Thanks to VIGIL, we could run 1,000 simulations of $CO_2$ gas dispersion in the area, with each simulation representing the 24 hours-long dispersion on a day randomly sampled from the period 01/01/1993 – 01/01/2023. The meteorological data retrieved by VIGIL from the ERA5 dataset (Hersbach et al., 2018a, b) were used to compute the wind field at high resolution by means of DIAGNO. Then, for each simulation day, by means of the new capabilities of VIGIL v1.3.7, we:

- varied the gas emission rate by randomly sampling it for each simulation from a normal distribution with mean 23.1 kg s$^{-1}$ and standard deviation 5.8 kg s$^{-1}$ according to Chiodini et al. (2010);
- evaluated the daily-averaged Richardson number (hence, neglecting possible intra-daily variations of the $Ri$ number) at the source was evaluated and, based on its value, carried out 24 hours-long simulations with a 1 hour time step with DISGAS or TWODEE-2. In this way we were not forced to focus on a specific scenario (e.g., the no-wind scenario of Chiodini et al. (2010), namely, to use one of the two models for all the simulations or to manually select the model to use for each day.
- Finally, the post-processing capabilities of VIGIL allowed us to produce hazard maps, hazard curves and persistence maps, which highlighted the occurrence of potentially dangerous concentrations of $CO_2$ at low levels in the atmosphere at non-negligible likelihoods. This is not a surprise, since fatalities occurred in the past and the main W-E valley in the considered domain is characterized by the lack of vegetation, which indicates the recurrent occurrence of the cold $CO_2$ gas stream. We could also obtain preliminary indications on the $H_2S$ hazard in the area, which should not be discarded and should be further tackled in future studies.

We need to stress that this study presents a partial hazard analysis showcasing the new capabilities of VIGIL. A more quantitative hazard assessment, which accounts also for other uncertainties (e.g., the lack of a recent gas emission source characterization due to the danger and difficulties of field surveys, the use of meteorological data from reanalyses in a fast evolving climate, etc.), is outside the scope of this manuscript. However, as recorded by the impacts on the local vegetation and historical chronicles, the Mefite area has been characterized by stable emission rates, similar to those dating back to the Roman era. Therefore, we do not expect a significant variation outside the range used in our work, which, in any case, fully include the statistical uncertainties estimated after the campaigns by Chiodini et al. (2010). Future developments of VIGIL will allow better treating the uncertainty of the source (both in the location and strength) and assessing the probability of death

depending on exposure duration, following the approach of Folch et al. (2017), who computed the percentage of human fatalities based on a probability density that depends on concentration thresholds and exposure times of the selected gas specie. Furthermore, we plan to use a more up-to-date wind processor that enables time steps shorter than one hour, which is the

495 current limit due to the use of DIAGNO. This in turn will allow improving the analysis of the impact of gas species concentrations on human health, since some of the concentration thresholds are related to exposure time of few minutes. In this way, we should also be able to produce more sophisticated probabilistic outputs of the impact of human health.

**Appendix A. Effect of the computational domain resolution on the modelling outputs.**

500 In order to check the effect of the spatial resolution of the discretized computational domain on the modelling output, we carried out test simulations in which we varied the spatial resolution. Specifically, we selected the scenario simulation carried out at $Ri = 0.438$ with 3 m resolution in the $Ri$-dependency test of Section 5 and carried out two further simulations with spatial resolutions of 1.5 and 6 m, respectively. We then calculated the $RMSE$ between the solution at the highest resolution (1.5 m) and the other two solutions (3 and 6 m) with both DISGAS and TWODEE:

505

$$RMSE = \sqrt{\frac{\Sigma_{i=1}^{N}\left(C_{i,res} - C_{i,res\_1.5m}\right)^2}{N}} \tag{A1}$$

Subsequently, we calculated the $MV$ of the concentration obtained with a resolution of 1.5 m:

$$MV = \frac{\Sigma_{i=1}^{N} C_{i,res\_1.5m}}{N} - C_{background} \tag{A2}$$

510

Results for DISGAS and TWODEE-2 are shown in Table A1 and A2, respectively:

| resolution | *RMSE* All domain [ppm] | *MV* All domain [ppm] | *RMSE/MV* [%] | *RMSE* Reduced domain [ppm] | *MV* Reduced domain [ppm] | *RMSE/MV* [%] |
|---|---|---|---|---|---|---|
| 3 m vs 1.5 m | 1.07 | 1223.77 | 0.09 | 3.15 | 15691.99 | 0.02 |
| 6 m vs 1.5 m | 4.73 | 1230.35 | 0.38 | 11.91 | 9232.30 | 0.13 |

**Table A1. Results of RMSE calculations for the resolution-dependency test for the DISGAS case.**

| resolution | *RMSE* All domain [ppm] | *MV* All domain [ppm] | *RMSE/MV* [%] | *RMSE* Reduced domain [ppm] | *MV* Reduced domain [ppm] | *RMSE/MV* [%] |
|---|---|---|---|---|---|---|
| 3 m vs 1.5 m | 44.53 | 1915.58 | 2.32 | 187.22 | 26863.64 | 0.70 |
| 6 m vs 1.5 m | 98.18 | 1920.91 | 5.11 | 286.52 | 15037.50 | 1.91 |

**Table A2. Results of RMSE calculations for the resolution-dependency test for the TWODEE2 case.**

Results show that, with the current resolution, the effect of a further refinement is negligible, in particular for DISGAS, which could be expected since DISGAS outputs are less affected by the topography than TWODEE-2.

**Code availability**

VIGIL v1.3.7, with which the PHA was carried out, is available at https://github.com/BritishGeologicalSurvey/VIGIL/releases/tag/v1.3.7.

**Data availability**

All the data and instructions necessary to replicate the presented PHA are available at https://zenodo.org/doi/10.5281/zenodo.10154599. The repository also includes all the probabilistic outputs generated by VIGIL.

**Author contribution**

FD contributed to the conceptualization, data curation, formal analysis, simulations, software development, visualization and writing. AC and GC contributed to the conceptualization, formal analysis and writing.

**Competing interests**

The authors declare that they have no conflict of interest.

**Acknowledgements**

Hersbach et al., 2018 was downloaded from the Copernicus Climate Change Service (C3S) (2023). The results contain modified Copernicus Climate Change Service information 2020. Neither the European Commission nor ECMWF is responsible for any use that may be made of the Copernicus information or data it contains. This work is published with permission of the Executive Director of British Geological Survey (UKRI). This study was carried out within the RETURN

Extended Partnership and received funding from the European Union Next-GenerationEU (National Recovery and Resilience Plan – NRRP, Mission 4, Component 2, Investment 1.3 – D.D. 1243 2/8/2022, PE0000005). We thank the Istituto Nazionale di Geofisica e Vulcanologia, Italy, grant "Progetto INGV Pianeta Dinamico - FURTHER" (code CUP D53J19000170001) funded by Italian Ministry MIUR ("Fondo Finalizzato al rilancio degli investimenti delle amministrazioni centrali dello Stato e allo sviluppo del Paese", legge 145/2018).. The computational work has been executed on the IT resources of the ReCaS-Bari data center, which have been made available by the following projects financed by the MIUR (Italian Ministry for Education, University and Research): ReCaS (Azione I - Interventi di rafforzamento strutturale, PONa3_00052, Avviso 254/Ric) and PRISMA (Asse II - Sostegno all'innovazione, PON04a2_A), within the "PON Ricerca e Competitività 2007-2013" program, and DHTCS (now IPCEI-HPCBDA, Avviso D. D. n. 424 del 28.02.2018), IBiSCo (PIR01_00011), CNRBiOmics (PIR01_00017) and LifeWatchPLUS (PIR-01_00028) within the "PON Ricerca e Innovazione 2014-2020" program Azione II.1, CUP I66C18000100006, ICSC and TeRABIT within the "PNRR – Avviso n 3264 per il "Rafforzamento e creazione di Infrastrutture di Ricerca", Missione 4, "Istruzione e Ricerca"". We would like to thank Dr. Giovanni Macedonio for the very fruitful discussions and for helping improving the DISGAS and TWODEE-2 codes.

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
