# Peer review of "Probabilistic hazard analysis of the gas emission of Mefite d'Ansanto, Southern Italy"

_EGUsphere, 2023_

## Author Comment (AC1)

We would like to thank the reviewer for his thorough review, which will help us improving the quality and the scientific soundness of our work.

In the following we reply to the reviewer's points and requests, which are here cited in italics.

*I would suggest, however, a better clarification of the newness of the article, since it is not so clear. Is there any step of the analyses that is new in the current study or it is just the first time is applied in this degassing area? All these procedures integrating different tools (DISGAS, TWODEE-2, DIAGNO) were already applied before? This should be better clarified.*

We agree with the reviewer that the novelty of this work should be better emphasized. In the reviewed version we will do this by further emphasizing in the abstract, introduction, discussion and conclusion that:

- Thanks to the latest version of VIGIL, we could carry out a more robust hazard assessment without imposing an a priori scenario, considering the natural variability of meteorological conditions and applying the most appropriate transport model (i.e., dense gas flow or dilute gas transport).
- Unlike the original work of Chiodini et al. (2010), who considered a single dense gas flow scenario with very little wind, we explored the effects of the full meteorological variability using the appropriate gas transport model, without imposing a pre-determined scenario. Such an approach allowed us also to capture the seasonal control on the hazard assessment.

*Authors estimate probability maps not only for the CO2, but decided also to include the H2S. Was this approach already applied in other sites? Considering that CH4 may also be hazardous in certain concentrations and that the CH4 concentration in the emission is also high (Table 1 from the article), did authors consider to apply the methodology also to the CH4?*

We are successfully applying this approach of converting among gas species in other sites of interest. To our knowledge, this modelling approach has never been applied by other authors in other sites.

From Table 1 one can estimate the molar ratio between CH4 and CO2; this is the parameter used by VIGIL to obtain an estimate of the concentration of a secondary gas specie (CH4) starting from the concentration of the main gas specie (in our case, CO2). This molar ratio is lower than the molar ratio between H2S and CO2 that we used in our work to produce the outputs shown in Figure 8. This implies that the predicted CH4 concentrations would be lower than the H2S concentrations of Figure 8a, which represent our worst case scenario since it is a map of H2S concentration at 5% exceedance probability. In the worst case scenario, then, we could expect maximum concentrations of CH4 in the region of few hundreds ppm. It is difficult to find Threshold Exposure Limits (TLV) for CH4, however some sources report a minimum TLV of 1000 ppm over 8 hours by NIOSH (well above what we would expect if we applied CH4 calculations with VIGIL); the second limit generally reported is 500,000 ppm, which would cause asphyxia; in the between there are the concentrations that would cause explosions (50,000 – 150,000) (https://www1.agric.gov.ab.ca/$department/deptdocs.nsf/all/agdex9038/$file/729-2.pdf?OpenElement&fbclid=IwAR1kqzolBzj46xk1fyTuQULkl3dsQiYdDI8LlIcnVLbVhEPQ3

3HIycI7v_c, https://pubs.geoscienceworld.org/eg/article-abstract/22/3/85/138156/Does-methane-pose-significant-health-and-public, https://journals.lww.com/epidem/fulltext/2011/01001/methane_and_natural_gas_exposure_limits.771.aspx). For these reasons, we do not think that it is worth including CH4 in our analysis for the area.

*Chiodini et al. (2010) applied the TWODEE-2 code to the same dataset at Mefite. Authors should better discuss what are the major differences obtained between the two different studies especially when DISGAS is applied to similar wind conditions as the ones applied in the previous study.*

As already mentioned above, Chiodini et al. (2010) considered a specific dense gas flow scenario with very little wind, whilst we conducted a scenario-independent hazard modelling with considerations on the seasonal control. In the revised version we will clarify this point.

**Some more general comments:**

*Section 1. I suggest authors to review the references used for the CO2 thresholds and exposure time. Authors refer in one case to an Italian reference (Settimo et al., 2022) together with some studies that do not focus on these thresholds but use them based on other literature. I suggest one of the options: or authors include the references and add "and references therein", or, in a better way, use more fundamental and specific literature on these thresholds, impacts and exposure time. Some recommendations are: Blong, 1984; Wong, 1996; IVHHN, NIOSH - https://stacks.cdc.gov/view/cdc/19367 (line 21 – page 1). I also suggest checking the references sequence in the text. Some of them do not appear with the chronological sequence.*

In Section 4.1 we indeed used "Settimo et al. (2022) and references therein" but we are happy to use the references you suggested in this section and the Introduction.

*Section 2. On the characterization of the area I suggest authors to better characterize the area including mention that the emissions are cold and the fluxes previously estimated by Rogie et al. (2000) and Chiodini et al. (2010). For instance, Rogie et al. (2000) report that CO2 concentrations > 30 vol.% were measured at an height higher than 2 m. This is an interesting aspect to recall in the discussion to compare with the results obtained in the current study.*

The cold sources are mentioned in line 36, 374, 411 and in Section 4.1 we specified the temperature used in the modelling of the source. However, we agree with the reviewer that this should be mentioned in this section too, since it deals with the emissions' description. Furthermore, since this is a hazard modelling work, we chose to focus on the outputs at 2 m above the ground, which is the height where human beings breathe. Given the very high concentrations at 2 m, we can safely expect high concentrations also at higher levels as observed by Rogie et al.

*Section 4. Tables 2 and 3 in my opinion need to be improved. Another column should be added in both tables to mention the maximum recommended exposure time for each of the thresholds. I would split the effects and the exposure times. Then, the last column would report the "tested exposure time in the current study". Otherwise it seems that humans can be exposed to 10 vol.% during 1 hour (Table 1) and this is not true and could even be*

*lethal. In fact, in Table 2 authors need to add that in this last threshold (100 000 = 10 vol.%) death can also be one of the effects. Same comment for the 500 ppm associated to the H2S. Still in what concerns Table 2, I suggest authors to use specific literature (NIOSH, Blong, Wong, as mentioned above, and complete the symptoms per threshold considered, since there is lack of symptoms in the table). The IDLH mentioned in the text (line 259, page 10) for the H2S also exists for the CO2 and should be mentioned (and added in the table). I suggest to review this section based on additional literature.*

We agree with the reviewer and we will implement the suggested modifications.

*Section 5. The discussion and evaluation of the hazard considering different seasons is a very interesting contribution that should be applied in other degassing areas. Nevertheless, it is important that future studies attempt to couple seasonal degassing maps with the meteorological data associated with the different seasons. Several studies on degassing areas showed that CO2 is usually higher during winter comparing to summer, and for this reason the evaluation of this coupled (and eventually contradictory) effect will be very interesting. Is this the first study that reports these seasonal maps? If yes, this is not mentioned in the abstract and it should.*

We will further emphasize these findings in the abstract. Additionally we will further explain that our outputs, specifically the concentration maps, are 24 time-averaged. In our view this is one of the reasons why the effect of winds, which is evident in the seasonal control, dominates over other effects that may dominate in specific time windows of a day. Specifically, the effect of temperature inversion, which can occur under stable conditions especially in winter and enhances the gas concentration at the lowest levels, plays a role during the night and early morning.

*The are **some general technical comments** that I would like to call the attention, namely the need to control all the figures and tables that do not appear correctly in the text. This needs to be carefully checked:*

We will check all the technical points listed below.

- *For instance, in line 29 – page 1, authors refer Figure 1, but this sentence refers to Figure 2b.*
- *Line 78, page 3 – the links to the webpages of NIOSH could be added as references (e.g. NIOSH, 2019), and then the links appear in the references list.*
- *Line 85, page 3 – authors should add the units also for the isotopes in the table.*
- *Figure 1 was redrawn from Chiodini et al. (2010). Looking at the literature, there are several geophysical studies (e.g. https://www.mdpi.com/1424-8220/23/3/1630; https://pubs.geoscienceworld.org/ssa/bssa/article-abstract/113/3/1102/620681/Hydrothermal-Seismic-Tremor-in-a-Wide-Frequency?redirectedFrom=fulltext) that were recently developed and I wonder if they could be used to improve the scheme showed. Authors should check. Anyway, remember that Figures 1 and 2 need to be checked as they are wrongly referenced in the text.*
- *Line 97, page 4 – check the chronological sequence of the references.*
- *Line 204, page 8 – when appears "figure 2" should appear "Figure 2".*
- *Figure 2 – I think Figure 2 could be improved by inserting in Figure 2b a square with the location of Figure 2c.*

- *Line 221, page 9 – check what figure should be mentioned in the text. It does not refer to Figure 1.*
- *Line 245, page 9 – the number of the tables also need to be checked along the text. I believe that authors refer to "Table 2" and not "Table 1" as it is written.*
- *Table 3 – there is a reference missing in the first line.*
- *Line 285, page 13 – I think authors meant "winds blowing" instead of "finds blowing". Check.*
- *Lines 294 and 295, page 14 – I could not see in the figure the statement "is also not negligible… along the main". Maybe I misunderstood the sentence, but please check.*
- *Line 307, page 14 – Table 4 instead of Table 3.*
- *Line 318, page 14 – I wonder why authors decided to check the exceedance probability of 16%, and not any other percentage? What was the criteria?*
- *Line 382, page 21 – authors mention that certain concentrations of $CO_2$ may be very dangerous for the human health, but I would even add for the "human life", as in certain concentrations $CO_2$ is lethal and it was even reported casualties in the area.*
- *Line 404, page 22 – "was evaluated" is repeated in the sentence.*

---

## Author Comment (AC2)

We would like to thank the reviewer for his thorough review, which will help us improving the quality and the scientific soundness of our work.

In the following we reply to the reviewer's points and requests, which are here reported in italics.

*The core models utilised by VIGIL are DISGAS and TWODEE. DISGAS is an advection-diffusion model that transports a passive scalar emitted from a steady source within a prescribed wind field, refined using the DIAGNO model. It disregards gravitational effects. TWODEE, on the other hand, is a shallow water model for transporting a gravity current that interacts with prescribed wind and topography. DISGAS is appropriate for diluted flows and strong wind conditions (Richardson number, Ri, much less than 1), whereas TWODEE is used when gravitational effects dominate (Ri much greater than 1). Intermediate regimes, which can exhibit behaviours divergent from these two models, are more complex. While the authors correctly introduce and discuss this point, they do not quantify the epistemic uncertainty introduced by these approximations. Comparing the two models in intermediate regimes can help quantify this uncertainty. The authors use DISGAS when Ri < 0.25 and TWODEE when Ri > 0.25. I recommend comparing CO2 gas concentrations obtained with TWODEE and DISGAS for a scenario with Ri ~ 0.25 and a wind field from W-SW to quantify the differences between the two approaches under challenging conditions*

*This comparison is crucial also because the persistence values along the valley shown in Figure 4 likely result mainly from TWODEE simulations. The Richardson number, used to decide which model to apply, is calculated at the source. However, dilution due to entrainment and turbulent diffusion tends to decrease the local Richardson number, making the current lighter. Additionally, the Richardson number is averaged over a whole day, meaning some hours with Ri < 0.25 are modelled with TWODEE, which stretches TWODEE's capabilities in diluted regimes. I recommend adding a comment in the main text to describe these approximations.*

The reviewer is right about the possible change of regime as the gas flow dilutes downstream, although the domain under analysis is rather small. To overcome this issue, one should use a more complex model which handles both regimes, which is computationally much more demanding than DISGAS and TWODEE2. Such a model would be significantly more demanding in terms of computational resources, hence making it difficult to apply for hazard studies like the one here presented. However, we will include a statement about this limitation in the main text.

Concerning the approximation used by VIGIL in the intermediate regime, to answer to the reviewer criticism, we use TWODEE2 for all the cases in which Ri > 0.25. This is a cautious approach from hazard assessment point of view, since TWODEE2 generally results in higher near surface concentrations. First, the number of simulations in this regime is 277, which corresponds to 27.70%; therefore, we agreed that an analysis of the outputs produced by the two models in this intermediate regime was needed. Specifically, we selected three simulations with the following values of Ri, chosen to be equally spaced between 0.25 and 1: 0.438, 0.625 and 0.812. For each Ri value, we conducted the simulation using both DISGAS and TWODEE. From the outputs, with a Python code we calculated the RMSE between the DISGAS and the TWODE2 solution in the domain for each vertical level and each time step, then we calculated the average of all the RMSEs for each Ri case.

$$RMSE = \sqrt{\frac{\sum_{i=1}^{N}\left(C_{i,DISGAS} - C_{i,TWODEE}\right)^2}{N}}$$

Results are summarized in Table 1 and show that the error is negligible from hazard assessment point of view, especially for $CO_2$, since it is lower than the background value and the typical maximum concentrations easily overcome 1000-10000 ppm. To be further conservative, we considered only the part of the domain where at least one of the two models computed concentration values above the background concentration used in the simulations (400 ppm). In this case we obtained larger RMSE values, but still not significant.

Nonetheless, in future versions of VIGIL we will improve the way VIGIL automatically handles this intermediate regime, e.g., by introducing the option to calculate a Ri-weighted output from the DISGAS and TWODEE2 outputs.

*Table 1. Results of RMSE calculations for the Richardson-dependency test.*

| Ri | RMSE All domain [ppm] | RMSE Reduced domain [ppm] |
|---|---|---|
| 0.438 | 54.71 | 102.38 |
| 0.625 | 60.84 | 159.56 |
| 0.812 | 77.82 | 155.10 |

*Another critical aspect needing better description is the effect of turbulent diffusion. In advection-diffusion models like DISGAS, turbulent diffusion is a key parameter determining how much the flow dilutes and how gas concentration decreases with distance. This parameter is also important in TWODEE2, alongside entrainment. The authors should explicitly explain how these parameters are calculated in the two models and how turbulent diffusion depends on mesh resolution. I suggest adding a map (even in the supplementary material) showing the depth-averaged horizontal turbulent diffusion used by both models for scenarios with median, strong, and weak wind conditions, using the paper's resolution (3 m) and a refined resolution (1.5 m).*

*Simulations with refined resolutions are also essential for quantifying the epistemic uncertainty due to numerical approximations in this specific application. Including the effect of resolution in a supplementary figure would be beneficial.*

Even if the purpose of this paper is to use DISGAS and TWODEE2 to calculate the gas hazard and not to validate these models, whose details have already been published in a few papers reported in the reference list, we will include further details on the turbulent diffusion parameterization used in DISGAS. TWODEE2 only allows controlling the numerical diffusion with a coefficient, which was left to the default value of 0.2; in our opinion there is no need to go into this detail in this manuscript for TWODEE2. We believe that a thorough analysis of the turbulent (or numerical) diffusion modelling in DISGAS and TWODEE2 is outside the scope of this paper. However, on a selected simulation day (the one at Ri = 0.438 of the test mentioned above), we carried out two further simulations with spatial resolutions of 1.5 and 6 m, respectively. We then calculated the RMSE between the solution at the highest resolution (1.5 m) and the other two solutions (3 and 6 m):

$$RMSE = \sqrt{\frac{\sum_{i=1}^{N}\left(C_{i,res} - C_{i,res\_1.5m}\right)^2}{N}}$$

Results of this analysis are summarized in Table 2:

*Table 2. Results of RMSE calculations for the resolution-dependency test.*

| | DISGAS | | TWODEE2 | |
|---|---|---|---|---|
| resolution | RMSE All domain [ppm] | RMSE Reduced domain [ppm] | RMSE All domain [ppm] | RMSE Reduced domain [ppm] |
| 3 m vs 1.5 m | 12.02 | 33.10 | 64.79 | 267.02 |
| 6 m vs 1.5 m | 23.09 | 58.13 | 130.73 | 361.82 |

Results show that overall, the RMSE obtained by decreasing the resolution increases but, although not negligible, is not significant. RMSE are higher with TWODEE2, but so are the maximum concentrations, therefore the two models behave similarly. Moreover, that stronger sensitivity of the results of TWODEE2 are likely due to the topographic control and not to the parameterization of the turbulence. Also, in this case to be further conservative we considered only the part of the domain where at least one of the two models computed concentration values above the background concentration used in the simulations (400 ppm). In this case we obtained larger RMSE values, but still below the atmospheric background value.

*All parameters used by the two models should be explicitly listed in a table to help readers understand the modelled conditions without needing to download the database and delve into the model configuration.*

This is not possible as many of the parameters depend on the meteorological conditions, hence they change across the 1000 simulations. This is why we provided all the input data in the Zenodo repository.

*Meteorological conditions used in the simulations should be presented more clearly. The spatial resolution is 30 km, correct? Where is the center of the cell used for this specific application located? What is the temporal resolution (I assume it is 1 hour)? I recommend including this information in Section 3.2.1. Additionally, I suggest adding a figure (even in the supplementary material) showing an example of the vertical profiles downloaded from the cited datasets up to the height of the numerical domain. Highlight the value used for the quantification of the Richardson number and the Monin-Obukhov length scale, as well as the value used to draw the wind speed distribution shown in Figure 7 (at 10 m above the ground). How much variability does DIAGNO introduce to the original wind profile? I recommend showing the median and standard deviation of the vertical profiles obtained by applying DIAGNO to the example meteorological condition in the same figure.*

We have improved the description of the approach used by VIGIL in Section 3.2.1; specifically, we have explained that the ERA5 data that are the nearest to the computational domain centre are retrieved and then interpolated in the centre of the

computational domain. Furthermore, we have clarified that the meteorological data are at 1 hour resolution, as in the original ERA5 dataset. Since the Richardson number is calculated at the source, the Richardson number is calculated using meteorological data calculated by DIAGNO at 10 m above the ground. As far as the Monin-Obukhov length scale is concerned, this is calculated internally by DISGAS and TWODEE2 using the temperature at 2 m above the ground and at the soil in the centre of the domain coming from the ERA5 dataset.

We think we need to clarify the meaning of Figure 7. It is the domain-averaged wind speed at 10 m above the ground, so there is no information on the wind vertical profiles here. We have made it clearer in the manuscript. We think there is no need to show the vertical profiles of the ERA5 dataset and DIAGNO for the simple reason that in DIAGNO we restricted our analysis up to 500 m above the ground (which is what needed for these applications). Within the first 500 m above the ground one has only a couple of points in the ERA5 dataset, making a comparison of the vertical profiles hard and, in our opinion, not meaningful.

*It is unclear to me how many simulations are performed, how the source distribution is sampled, and how much computational time is required. Specifically, it appears that 24 simulations for each of the 1000 sample days are performed to account for the variability of meteorological conditions, resulting in a total of 24,000 1-hour simulations with fixed source conditions. How is the source variability taken into account? If the normal distribution introduced in the manuscript is sampled with, say, 10 points, the total number of simulations would be 240,000, correct? Please clarify these points in the main text.*

We run 24 hours-long 1000 simulations, we have made it clearer in the manuscript.

*Specific comments*

*Web links: Numerous web links are present in the main text. Is it possible to move them to a specific reference section with the date of last access?*

To our knowledge there are no prescriptions to authors about the URLs. Anyway we will cut several URLs by including new references.

*Zenodo Dataset: I was unable to access the Zenodo dataset.*

Indeed there were problems with the link. We have restored it.

*Section 2: The source data are from several years ago (Chiodini et al. 2010 and Rogie et al. 2000). Please add a comment on the expected variability at the source after 14 years (for the mass flow rate) and 24 years (for the composition).*

We agree that a new measurement campaign would be useful, but it is quite difficult and dangerous since the high gas concentrations in the area. However, as recorded by the impacts on the local vegetation and historical chronicles, the Mefite area has been characterized by stable emission rates, similar to those dating back to the Roman era. Therefore, we do not expect a significant variation outside the range used in our work, which, in any case, fully include the statistical uncertainties estimated after the campaings by Chiodini et al. (2010).

*Line 113: Is the momentum coupling one-way or two-way? Please specify explicitly.*

It's one way, we clarified this in the manuscript.

*Section 3: Please include a comment about the potential chemical reactions affecting H2S during its transport and their time scales.*

The main sink of H2S in the atmosphere is the reaction with OH radicals (e.g., Watts, 2000), other minor sinks can be found on a local scale, during and subsequent to rainfall events (Kristmannsdottir et al., 2000, Thorsteinsson et al., 2013) or under the action of lakes, soils, and vegetation (Bussotti et al., 1997; Cihacek and Bremner, 1990). However, these interactions typically do not have a first order control. For example, Olafsdottir et al. (2014) conducted ad hoc measurement campaigns in Iceland showing that the depletion of H2S from the atmosphere is insignificant compared to the emissions within a 35 km distance from the sources. Neglecting such reactions could imply an overestimation of the H2S concentration, probably not significant for our restricted domain. We clarified this point in the text.

*Lines 147-151: The difference between Forecast and Reanalysis mode is not perfectly clear to me. Why can't the ERA5 dataset be used in Forecast mode and NCEP in Reanalysis mode? Please clarify the differences between these two datasets.*

NCEP is a dataset that includes GFS daily forecasts for the next 384 hours. ERA5 is a reanalysis, therefore it cannot be used for forecasts (the corresponding dataset is called IFS: https://www.ecmwf.int/en/forecasts/documentation-and-support/changes-ecmwf-model ).

*Line 176: ECDF is not defined when first introduced. It is defined later at line 186.*

We fixed this in the manuscript.

*Section 4.1: The height of the numerical domain and the vertical size of the cells are not reported.*

This is because there is no vertical domain and discretization along the vertical direction in TWODEE2, being a shallow layer model (apart from vertical levels used to print the interpolated outputs), whilst there is in DISGAS and DIAGNO and they don't need necessarily to coincide.

*Figure 3: Please add a box highlighting the emission area.*

We fixed this in the manuscript.

*Figures 3, 4, 5, 6, 8: Please indicate, for each figure, how many scenarios come from TWODEE2 and how many from DISGAS simulations. This information is essential to understand the regimes producing hazardous conditions.*

This was clearly stated in lines 228-229, in our opinion there is no need to repeat this information in the figure captions and descriptions.

*Line 285: Change "finds blowing" to "winds blowing."*

We fixed this in the manuscript.

*Figure 5: It is not clear whether the curves represent 24-hour averages or hourly results.*

It's 24-hour averages, we clarified this in the manuscript.

*Figure 4 and Comments in the Text: The method for calculating the persistence maps is not completely clear to me. For example, when you say CO2 persistence > 5000 ppm for 8 hours, does this mean selecting (cell by cell) all the 24-hour scenarios with 8 consecutive hours satisfying the condition? If the condition is satisfied for 4 hours, then concentration decreases for 1 hour and then increases again for 4 hours, do you keep or discard the scenario?*

The persistence is defined as the probability to overcome a certain concentration threshold (in your example, 5000 ppm) for a duration (consecutive hours in this application) of at least N hours (in your example, 8 hours). In your example, the simulation would be discarded in the calculation of the persistence.

*Figure 7: Link the figure to the discussion at line 225.*

This request is not clear.

*Line 387: It is unclear how a lower estimate in a probabilistic hazard map (Figure 4def) can be described as "safe."*

We agree, the term "safe" is misleading. We removed it.

*Lines 402-403: It is not specified how many points were used to sample the source variability.*

For each simulation, a value of the source emission rate is sampled from the normal distribution. We made it clearer in the manuscript.

*Lines 404-406: State explicitly that this approach disregards intra-day variability of the Richardson number.*

We fixed this in the manuscript.

---

## Referee Report (RR1)

*The core models utilised by VIGIL are DISGAS and TWODEE. DISGAS is an advection-diffusion model that transports a passive scalar emitted from a steady source within a prescribed wind field, refined using the DIAGNO model. It disregards gravitational effects. TWODEE, on the other hand, is a shallow water model for transporting a gravity current that interacts with prescribed wind and topography. DISGAS is appropriate for diluted flows and strong wind conditions (Richardson number, Ri, much less than 1), whereas TWODEE is used when gravitational effects dominate (Ri much greater than 1). Intermediate regimes, which can exhibit behaviours divergent from these two models, are more complex. While the authors correctly introduce and discuss this point, they do not quantify the epistemic uncertainty introduced by these approximations. Comparing the two models in intermediate regimes can help quantify this uncertainty. The authors use DISGAS when Ri < 0.25 and TWODEE when Ri > 0.25. I recommend comparing $CO_2$ gas concentrations obtained with TWODEE and DISGAS for a scenario with Ri ~ 0.25 and a wind field from W-SW to quantify the differences between the two approaches under challenging conditions*

*This comparison is crucial also because the persistence values along the valley shown in Figure 4 likely result mainly from TWODEE simulations. The Richardson number, used to decide which model to apply, is calculated at the source. However, dilution due to entrainment and turbulent diffusion tends to decrease the local Richardson number, making the current lighter. Additionally, the Richardson number is averaged over a whole day, meaning some hours with Ri < 0.25 are modelled with TWODEE, which stretches TWODEE's capabilities in diluted regimes. I recommend adding a comment in the main text to describe these approximations.*

The reviewer is right about the possible change of regime as the gas flow dilutes downstream, although the domain under analysis is rather small.

*It's unclear what the domain is being described as "small" in relation to. The significance of a regime change is determined by the dilution rate, and no domain is inherently "small": its size depends on the specific variability scale of the phenomena being modeled. In my view, the possible change of regime as the gas flow dilutes downstream, along with the model's approximations, should be explicitly addressed in the description of the methodology to provide clarity on how the domain size relates to the physical processes being studied.*

To overcome this issue, one should use a more complex model which handles both regimes, which is computationally much more demanding than DISGAS and TWODEE2. Such a model would be significantly more demanding in terms of computational resources, hence making it difficult to apply for hazard studies like the one here presented.

*I agree this point and, at my advice, a sentence like this one should be put in the text.*

Concerning the approximation used by VIGIL in the intermediate regime, to answer to the reviewer criticism, we use TWODEE2 for all the cases in which Ri > 0.25. This is a cautious approach from hazard assessment point of view, since TWODEE2 generally results in higher near surface concentrations. First, the number of simulations in this regime is 277, which corresponds to 27.70%; therefore, we agreed that an analysis of the outputs produced by the two models in this intermediate regime was needed.

*Ok.*

Specifically, we selected three simulations with the following values of Ri, chosen to be equally spaced between 0.25 and 1: 0.438, 0.625 and 0.812. For each Ri value, we conducted the simulation using both DISGAS and TWODEE. From the outputs, with a Python code we calculated the RMSE between the DISGAS and the TWODE2 solution in the domain for each vertical level and each time step, then we calculated the average of all the RMSEs for each Ri case.

$$RMSE = \sqrt{\frac{\sum_{i=1}^{N}(C_{i,DISGAS} - C_{i,TWODEE})^2}{N}}$$

Results are summarized in Table 1 and show that the error is negligible from hazard assessment point of view, especially for CO2, since it is lower than the background value and the typical maximum concentrations easily overcome 1000-10000 ppm. To be further conservative, we considered only the part of the domain where at least one of the two models computed concentration values above the background concentration used in the simulations (400 ppm). In this case we obtained larger RMSE values, but still not significant.

*It is not correct to conclude that an effect is insignificant by comparing RMSE values with maximum values. Instead, RMSE should be compared with the mean value above the background concentration across both the entire domain and the reduced domain. This mean value should be calculated as:*

$$MV = \frac{\sum_{i=1}^{N} C_{i,DISGAS}}{N} - C_{background}$$

*These values should be included in Table 5 of the revised manuscript as a reference point for the RMSE values provided. Furthermore, since the hazard is presented as a map rather than a spatial average, a map showing the difference between the two models in either the worst-case or a representative scenario is necessary. This will help illustrate the potential epistemic error in the hazard maps as a function of location. This difference map should be generated at the same height at which the hazard maps were extrapolated. Additionally, the manuscript should specify the height at which the MV and RMSE were calculated.*

Nonetheless, in future versions of VIGIL we will improve the way VIGIL automatically handles this intermediate regime, e.g., by introducing the option to calculate a Ri-weighted output from the DISGAS and TWODEE2 outputs.

We included these considerations in lines 197-200, 435-457,

*Table 1. Results of RMSE calculations for the Richardson-dependency test.*

| Ri | RMSE All domain [ppm] | RMSE Reduced domain [ppm] |
|---|---|---|
| 0.438 | 54.71 | 102.38 |
| 0.625 | 60.84 | 159.56 |
| 0.812 | 77.82 | 155.10 |

*Another critical aspect needing better description is the effect of turbulent diffusion. In advection-diffusion models like DISGAS, turbulent diffusion is a key parameter determining*

*how much the flow dilutes and how gas concentration decreases with distance. This parameter is also important in TWODEE2, alongside entrainment. The authors should explicitly explain how these parameters are calculated in the two models and how turbulent diffusion depends on mesh resolution. I suggest adding a map (even in the supplementary material) showing the depth-averaged horizontal turbulent diffusion used by both models for scenarios with median, strong, and weak wind conditions, using the paper's resolution (3 m) and a refined resolution (1.5 m).*

*Simulations with refined resolutions are also essential for quantifying the epistemic uncertainty due to numerical approximations in this specific application. Including the effect of resolution in a supplementary figure would be beneficial.*

Even if the purpose of this paper is to use DISGAS and TWODEE2 to calculate the gas hazard and not to validate these models, whose details have already been published in a few papers reported in the reference list, we included further details on the turbulent diffusion parameterization used in DISGAS in lines 128-132. TWODEE2 only allows controlling the numerical diffusion with a coefficient, which was left to the default value of 0.2; in our opinion there is no need to go into this detail in this manuscript for TWODEE2. We believe that a thorough analysis of the turbulent (or numerical) diffusion modelling in DISGAS and TWODEE2 is outside the scope of this paper. However, on a selected simulation day (the one at Ri = 0.438 of the test mentioned above), we carried out two further simulations with spatial resolutions of 1.5 and 6 m, respectively. We then calculated the RMSE between the solution at the highest resolution (1.5 m) and the other two solutions (3 and 6 m):

$$RMSE = \sqrt{\frac{\sum_{i=1}^{N}\left(C_{i,res} - C_{i,res\_1.5m}\right)^2}{N}}$$

Results of this analysis are summarized in Table 2:

*Table 2. Results of RMSE calculations for the resolution-dependency test.*

| | DISGAS | | TWODEE2 | |
| --- | --- | --- | --- | --- |
| resolution | RMSE All domain [ppm] | RMSE Reduced domain [ppm] | RMSE All domain [ppm] | RMSE Reduced domain [ppm] |
| 3 m vs 1.5 m | 12.02 | 33.10 | 64.79 | 267.02 |
| 6 m vs 1.5 m | 23.09 | 58.13 | 130.73 | 361.82 |

Results show that overall, the RMSE obtained by decreasing the resolution increases but, although not negligible, is not significant. RMSE are higher with TWODEE2, but so are the maximum concentrations, therefore the two models behave similarly. Moreover, that stronger sensitivity of the results of TWODEE2 are likely due to the topographic control and not to the parameterization of the turbulence. Also, in this case to be further conservative we considered only the part of the domain where at least one of the two models computed concentration values above the background concentration used in the simulations (400 ppm). In this case we obtained larger RMSE values, but still below the atmospheric background value.

*The same comment applies here as previously mentioned: the mean value (MV) and a representative difference map should be included and discussed in the paper. These additions will help provide a clearer understanding of the model's accuracy and potential epistemic errors across the domain.*

*All parameters used by the two models should be explicitly listed in a table to help readers understand the modelled conditions without needing to download the database and delve into the model configuration.*

This is not possible as many of the parameters depend on the meteorological conditions, hence they change across the 1000 simulations. This is why we provided all the input data in the Zenodo repository.

*Thank you for your clarification regarding the model's parameters. I understand that some of these parameters may vary significantly, and while this variability is important, I believe it would still be highly beneficial to present them in a table. For parameters with a wide range, simply reporting the range of variability should suffice. However, for parameters that are kept fixed, such as the numerical diffusion mentioned, it is crucial to include these explicitly in the manuscript. Providing a clear list of all parameters used in the simulation ensemble would greatly enhance the reader's ability to quickly understand the key assumptions and approximations underlying the model. This would also improve the transparency and reproducibility of the work.*

*Meteorological conditions used in the simulations should be presented more clearly. The spatial resolution is 30 km, correct? Where is the center of the cell used for this specific application located? What is the temporal resolution (I assume it is 1 hour)? I recommend*

*including this information in Section 3.2.1. Additionally, I suggest adding a figure (even in the supplementary material) showing an example of the vertical profiles downloaded from the cited datasets up to the height of the numerical domain. Highlight the value used for the quantification of the Richardson number and the Monin-Obukhov length scale, as well as the value used to draw the wind speed distribution shown in Figure 7 (at 10 m above the ground). How much variability does DIAGNO introduce to the original wind profile? I recommend showing the median and standard deviation of the vertical profiles obtained by applying DIAGNO to the example meteorological condition in the same figure.*

We improved the description of the approach used by VIGIL in Section 3.2.1; specifically, we explained that the ERA5 data that are the nearest to the computational domain centre are retrieved and then interpolated in the centre of the computational domain (line 177). Furthermore, we clarified that the meteorological data are at 1 hour resolution, as in the original ERA5 dataset (line 236). Since the Richardson number is calculated at the source, the Richardson number is calculated using meteorological data calculated by DIAGNO at 10 m above the ground. As far as the Monin-Obukhov length scale is concerned, this is calculated internally by DISGAS and TWODEE2 using the temperature at 2 m above the ground and at the soil in the centre of the domain coming from the ERA5 dataset.

We think we need to clarify the meaning of Figure 7. It is the domain-averaged wind speed at 10 m above the ground, so there is no information on the wind vertical profiles here. We made it clearer in the manuscript (lines 355-356). We think there is no need to show the vertical profiles of the ERA5 dataset and DIAGNO for the simple reason that in DIAGNO we restricted our analysis up to 500 m above the ground (which is what needed for these applications). Within the first 500 m above the ground one has only a couple of points in the ERA5 dataset, making a comparison of the vertical profiles hard and, in our opinion, not meaningful.

*OK. I suggest explicitly stating that the regime is selected by calculating the Richardson number using the wind field at 10 m above the ground. Additionally, details regarding the data used to evaluate the Monin-Obukhov length scale should be included in the manuscript to enhance transparency.*

*Regarding the vertical wind profiles, I understand that Figure 7 represents data at a fixed height and that the available meteorological data have relatively low vertical resolution. However, for this very reason, it is crucial to comment on the variability of the original dataset. For instance, if the dataset includes only two vertical points, it would be useful to illustrate this variability using Rose diagrams. This would help explain how the limited data are translated into the much more detailed wind field produced by the DIAGNO model.*

*To improve clarity, one option could be to present Rose diagrams side by side: one for the heights where meteorological data are available and another for the wind field at 10 m above ground level, as generated by DIAGNO. Additionally, I believe it would be helpful, if possible, to show an example of a vertical wind profile from the DIAGNO model, overlaid with the two original points from the ERA5 dataset. This would provide a clearer view of how DIAGNO refines the coarser input data into a more resolved wind profile.*

*It is unclear to me how many simulations are performed, how the source distribution is sampled, and how much computational time is required. Specifically, it appears that 24 simulations for each of the 1000 sample days are performed to account for the variability of meteorological conditions, resulting in a total of 24,000 1-hour simulations with fixed*

*source conditions. How is the source variability taken into account? If the normal distribution introduced in the manuscript is sampled with, say, 10 points, the total number of simulations would be 240,000, correct? Please clarify these points in the main text.*

We run 24 hours-long 1000 simulations, we made it clearer in the manuscript (lines 240-250, 469-470).

*OK. The fact that the gas emission rate is varied by randomly sampling it across the 1,000 simulations should be explicitly stated also in lines 240-250 to prevent any confusion. In its current form, Section 4.1 gives the impression that 1,000 days with different meteorological conditions were selected independently of the source conditions. However, as I now understand, this is not the case. Each day could potentially have a different source condition, randomly sampled as briefly described later in the conclusion. This raises a question about the stability of the results: if the same workflow were run again, different outcomes could emerge due to the random sampling. While I recognize the constraints imposed by computational resources and am comfortable assuming this effect is likely negligible, I believe this point should still be explicitly addressed in the manuscript. Clearly describing this aspect will help readers understand the inherent variability in the results and the model's robustness.*

*Specific comments*

*Web links: Numerous web links are present in the main text. Is it possible to move them to a specific reference section with the date of last access?*

To our knowledge there are no prescriptions to authors about the URLs. Anyway we cut several URLs by including new references.

*Zenodo Dataset: I was unable to access the Zenodo dataset.*

Indeed there were problems with the link. We restored it.

*OK*

*Section 2: The source data are from several years ago (Chiodini et al. 2010 and Rogie et al. 2000). Please add a comment on the expected variability at the source after 14 years (for the mass flow rate) and 24 years (for the composition).*

We agree that a new measurement campaign would be useful, but it is quite difficult and dangerous since the high gas concentrations in the area. However, as recorded by the impacts on the local vegetation and historical chronicles, the Mefite area has been characterized by stable emission rates, similar to those dating back to the Roman era. Therefore, we do not expect a significant variation outside the range used in our work, which, in any case, fully include the statistical uncertainties estimated after the campaings by Chiodini et al. (2010).

*OK. Please incorporate this reasoning into the manuscript for clarity.*

*Line 113: Is the momentum coupling one-way or two-way? Please specify explicitly.*

It's one way, we clarified this in the manuscript (lines 122-123).

*OK*

*Section 3: Please include a comment about the potential chemical reactions affecting H2S during its transport and their time scales.*

The main sink of H2S in the atmosphere is the reaction with OH radicals (e.g., Watts, 2000), other minor sinks can be found on a local scale, during and subsequent to rainfall events (Kristmannsdottir et al., 2000, Thorsteinsson et al., 2013) or under the action of lakes, soils, and vegetation (Bussotti et al., 1997; Cihacek and Bremner, 1990). However, these interactions typically do not have a first order control. For example, Olafsdottir et al. (2014) conducted ad hoc measurement campaigns in Iceland showing that the depletion of H2S from the atmosphere is insignificant compared to the emissions within a 35 km distance from the sources. Neglecting such reactions could imply an overestimation of the H2S concentration, probably not significant for our restricted domain. We clarified this point in the text (lines 255-264, 396-400).

*OK, thanks.*

*Lines 147-151: The difference between Forecast and Reanalysis mode is not perfectly clear to me. Why can't the ERA5 dataset be used in Forecast mode and NCEP in Reanalysis mode? Please clarify the differences between these two datasets.*

NCEP is a dataset that includes GFS daily forecasts for the next 384 hours. ERA5 is a reanalysis, therefore it cannot be used for forecasts (the corresponding dataset is called IFS: https://www.ecmwf.int/en/forecasts/documentation-and-support/changes-ecmwf-model ).

*OK*

*Line 176: ECDF is not defined when first introduced. It is defined later at line 186.*

We fixed this in the manuscript (line 193)

*OK*

*Section 4.1: The height of the numerical domain and the vertical size of the cells are not reported.*

This is because there is no vertical domain and discretization along the vertical direction in TWODEE2, being a shallow layer model (apart from vertical levels used to print the interpolated outputs), whilst there is in DISGAS and DIAGNO and they don't need necessarily to coincide. Anyway we provided necessary details in lines 243-244,

*Ok.*

*Figure 3: Please add a box highlighting the emission area.*

It is already shown in fig. 2c, we tried adding the box in the output figures and the resulting picture was not optimal in our opinion.

*OK*

*Figures 3, 4, 5, 6, 8: Please indicate, for each figure, how many scenarios come from TWODEE2 and how many from DISGAS simulations. This information is essential to understand the regimes producing hazardous conditions.*

This was clearly stated in lines 250-251, in our opinion there is no need to repeat this information in the figure captions and descriptions.

*OK.*

*Line 285: Change "finds blowing" to "winds blowing."*

We fixed this in the manuscript (line 327).

*OK*

*Figure 5: It is not clear whether the curves represent 24-hour averages or hourly results.*

It's 24-hour averages, we clarified this in the manuscript (line 358, 369).

*OK*

*Figure 4 and Comments in the Text: The method for calculating the persistence maps is not completely clear to me. For example, when you say $CO_2$ persistence > 5000 ppm for 8 hours, does this mean selecting (cell by cell) all the 24-hour scenarios with 8 consecutive hours satisfying the condition? If the condition is satisfied for 4 hours, then concentration decreases for 1 hour and then increases again for 4 hours, do you keep or discard the scenario?*

The persistence is defined as the probability to overcome a certain concentration threshold (in your example, 5000 ppm) for a duration (consecutive hours in this application) of at least N hours (in your example, 8 hours). In your example, the simulation would be discarded in the calculation of the persistence.

*OK.*

*Figure 7: Link the figure to the discussion at line 225.*

This request is not clear.

*I believe Figure 7 could be highly useful for readers attempting to understand how many scenarios fall into one regime or another. To enhance clarity, I suggest adding a reference to Figure 7 in support of the discussion around line 225. This would help readers better visualize the distribution of scenarios across different regimes. Additionally, I would like to point out that the reasoning around line 225 is based on the mean gas emission rate. Please ensure this is made explicit in the text to avoid any ambiguity.*

*Line 387: It is unclear how a lower estimate in a probabilistic hazard map (Figure 4def) can be described as "safe."*

We agree, the term "safe" is misleading. We removed it.

*OK*

*Lines 402-403: It is not specified how many points were used to sample the source variability.*

For each simulation, a value of the source emission rate is sampled from the normal distribution. We made it clearer in the manuscript (line 466).

*OK*

*Lines 404-406: State explicitly that this approach disregards intra-day variability of the Richardson number.*

We fixed this in the manuscript (line 468).

*OK.*

---

## Author Response (AR2)

Dear editor,

We would like to thank the reviewer for his additional comments that helped us to further improve the clarity of the manuscript, although we do not think that they needed neither a major revision to be addressed nor more than three months to be delivered. As we explain below, a few comments were relevant to clarify some points, while we found other comments/suggestions not fully relevant for the goal of the paper, that is a gas hazard analysis and not a fluid dynamic study on a specific test case, nor a validation of the simulation tools (DISGAS and TWODEE-2) used by VIGIL.

In the following we list the reviewer's comments that needed to be addressed in italics and our responses. When lines in the manuscript are reported, these refer to the track-changes version of the file.

*It's unclear what the domain is being described as "small" in relation to. The significance of a regime change is determined by the dilution rate, and no domain is inherently "small": its size depends on the specific variability scale of the phenomena being modeled. In my view, the possible change of regime as the gas flow dilutes downstream, along with the model's approximations, should be explicitly addressed in the description of the methodology to provide clarity on how the domain size relates to the physical processes being studied.*

*(…)*

*I agree this point and, at my advice, a sentence like this one should be put in the text.*

We added a few lines addressing this point (lines 195-201). We agree that the transport regime can change as the gas flow dilutes downstream, but as we explain in the answer to the next point, the use of one model (e.g., DISGAS) or the other (e.g., TWODEE-2) does not result in a significant discrepancy and, from a hazard assessment point of view, does not imply an underestimation of the hazard.

*It is not correct to conclude that an effect is insignificant by comparing RMSE values with maximum values. Instead, RMSE should be compared with the mean value above the background concentration across both the entire domain and the reduced domain. This mean value should be calculated as:*

$$MV = \frac{\sum_{i=1}^{N} C_{i,DISGAS}}{N} - C_{background}$$

*These values should be included in Table 5 of the revised manuscript as a reference point for the RMSE values provided. Furthermore, since the hazard is presented as a map rather than a spatial average, a map showing the difference between the two models in either the worst-case or a representative scenario is necessary. This will help illustrate the potential epistemic error in the hazard maps as a function of location. This difference map should be generated at the same height at which the hazard maps were extrapolated. Additionally, the manuscript should specify the height at which the MV and RMSE were calculated.*

*(…)*

*The same comment applies here as previously mentioned: the mean value (MV) and a representative difference map should be included and discussed in the paper. These additions will help provide a clearer understanding of the model's accuracy and potential epistemic errors across the domain.*

As suggested by the reviewer, we calculated the Mean Values (MV) for both tests in the following ways:

- For the Richardson number-dependency test (Table 1), for each tested Richardson number scenario, MV was calculated as suggested by the reviewer but, since the RMSE was calculated comparing the concentrations predicted with DISGAS and TWODEE2, MV was calculated by averaging the MVs obtained by each model.
- For the resolution-dependency tests (Table 2 and 3), for each test (3m vs 1.5 m and 6 m vs 1.5 m), MV was calculated using the concentration obtained with a resolution of 1.5 m.

In both cases, the background concentration was set to 400 ppm. Furthermore, both RMSE and MV were calculated across all the domain and taking all the output heights into account.

As it can be seen from Table 1, the ratios between RMSE and MV in all tested cases range from 0.02% to 5.11%, therefore we can conclude that both the error introduced by the scenario selection (dilute vs. dense gas) and the error introduced by the resolution selection is not significant. For the same reason, we do not deem necessary to introduce new maps in the manuscript that can make the reading heavy without providing useful information. However, we have now explained this point at lines 446-453 and modified Table 5 in the manuscript. Table 2 and 3 presented here, are now added in Appendix 1, with an explanation on the test and its results now included at lines 463-467.

*Table 1. Results of RMSE calculations for the Richardson-dependency test.*

| Ri | RMSE All domain [ppm] | MV All domain (ppm) | RMSE/MV [%] | RMSE Reduced domain [ppm] | MV Reduced domain [ppm] | RMSE/MV [%] |
|---|---|---|---|---|---|---|
| 0.438 | 54.71 | 1246.25 | 4.39 | 102.38 | 4360.19 | 2.35 |
| 0.625 | 60.84 | 1223.66 | 4.97 | 159.56 | 8051.21 | 1.98 |
| 0.812 | 77.82 | 2326.30 | 3.35 | 155.10 | 9576.85 | 1.62 |

*Table 2. Results of RMSE calculations for the resolution-dependency test for the DISGAS case*

| resolution | RMSE All domain [ppm] | MV All domain [ppm] | RMSE/MV [%] | RMSE Reduced domain [ppm] | MV Reduced domain [ppm] | RMSE/MV [%] |
|---|---|---|---|---|---|---|
| 3 m vs 1.5 m | 1.07 | 1223.77 | 0.09 | 3.15 | 15691.99 | 0.02 |
| 6 m vs 1.5 m | 4.73 | 1230.35 | 0.38 | 11.91 | 9232.30 | 0.13 |

*Table 3. Results of RMSE calculations for the resolution-dependency test for the TWODEE2 case*

| resolution | RMSE All domain [ppm] | MV All domain [ppm] | RMSE/MV [%] | RMSE Reduced domain [ppm] | MV Reduced domain [ppm] | RMSE/MV [%] |
|---|---|---|---|---|---|---|
| 3 m vs 1.5 m | 44.53 | 1915.58 | 2.32 | 187.22 | 26863.64 | 0.70 |
| 6 m vs 1.5 m | 98.18 | 1920.91 | 5.11 | 286.52 | 15037.50 | 1.91 |

*Thank you for your clarification regarding the model's parameters. I understand that some of these parameters may vary significantly, and while this variability is important, I believe it would still be highly beneficial to present them in a table. For parameters with a wide range, simply reporting the range of variability should suffice. However, for parameters that are kept fixed, such as the numerical diffusion mentioned, it is crucial to include these explicitly in the manuscript. Providing a clear list of all parameters used in the simulation ensemble would greatly enhance the reader's ability to quickly understand the key assumptions and approximations underlying the model. This would also improve the transparency and reproducibility of the work.*

We do not agree there is need for additional tables, as the few parameters that are fixed in DISGAS and TWODEE-2 simulations are clearly specified in the manuscript while the other that are varied are fully discussed or the proper literature reference provided (e.g., the turbulent diffusion coefficients for DISGAS, calculated as explained in lines 123-126). The main variables that were varied to capture their statistical variability were the wind field in the whole domain, based on the DIAGNO outputs initialized from ERA-5 reanalysis (for the details on how  DIAGNO works we properly cited the DWM User Guide that obviously cannot be summarized in a table or even in the article) and the emission rate, for which we explicitly said that the values were varied sampling from a normal distribution (lines 225-226), whose statistical parameters are reported clearly in the manuscript. Moreover, all the data, codes, input files, instructions to re-run the PHA are clearly explained in the manuscript and provided in the Zenodo repository. For these reasons we consider the reviewer comments that constantly question the transparency and reproducibility of the work unfounded.

*OK. I suggest explicitly stating that the regime is selected by calculating the Richardson number using the wind field at 10 m above the ground.*

Done in line 250.

*Additionally, details regarding the data used to evaluate the Monin-Obukhov length scale should be included in the manuscript to enhance transparency.*

Done in lines 132-136.

*Regarding the vertical wind profiles, I understand that Figure 7 represents data at a fixed height and that the available meteorological data have relatively low vertical resolution. However, for this very reason, it is crucial to comment on the variability of the original dataset. For instance, if the dataset includes only two vertical points, it would be useful to illustrate this variability using Rose diagrams. This would help explain how the limited data*

*are translated into the much more detailed wind field produced by the DIAGNO model. To improve clarity, one option could be to present Rose diagrams side by side: one for the heights where meteorological data are available and another for the wind field at 10 m above ground level, as generated by DIAGNO. Additionally, I believe it would be helpful, if possible, to show an example of a vertical wind profile from the DIAGNO model, overlaid with the two original points from the ERA5 dataset. This would provide a clearer view of how DIAGNO refines the coarser input data into a more resolved wind profile.*

We disagree with the reviewer on this comment and on the need to add further figures to our manuscript in addition to figure 7. As we explained above and in the text, we used ECMWF ERA5 reanalysis meteorological data, which resolution is too coarse to capture local variability and topographic effects, to initialize the mass consistent wind module DIAGNO, which provides the local high resolution wind field near the ground that is then used to run the gas transport model. The goal here is not to compare the wind field from large to local scale, for which the reader can refer to the DWM user guide, but to statistically capture its variability in order to build a probability map. From a statistical point view, it was demonstrated that, for hazard analysis scopes, the used dataset is not critical for capturing the aleatoric uncertainty (see for example Macedonio et al., 2016 analysed the effects of different dataset for the case dispersion of volcanic ash on a larger scale: Macedonio G., Costa A., Scollo S., Neri A. (2016) Effects of eruption source parameter variation and meteorological dataset on tephra fallout hazard assessment: an example from Vesuvius (Italy), J. Appl. Volcanol., 5, 5, 1-19, doi:10.1186/s13617-016-0045-2).

However, we added few lines to stress this point at 132-136.

*OK. The fact that the gas emission rate is varied by randomly sampling it across the 1,000 simulations should be explicitly stated also in lines 240-250 to prevent any confusion. In its current form, Section 4.1 gives the impression that 1,000 days with different meteorological conditions were selected independently of the source conditions. However, as I now understand, this is not the case. Each day could potentially have a different source condition, randomly sampled as briefly described later in the conclusion. This raises a question about the stability of the results: if the same workflow were run again, different outcomes could emerge due to the random sampling. While I recognize the constraints imposed by computational resources and am comfortable assuming this effect is likely negligible, I believe this point should still be explicitly addressed in the manuscript. Clearly describing this aspect will help readers understand the inherent variability in the results and the model's robustness.*

The emission rate was sampled randomly by a normal distribution as already stated in lines 225-226. We added a new sentence in lines 250-251 which we hope will make clearer that the emission rate was varied, within the normal distribution, for each of the sampled day.

*OK. Please incorporate this reasoning into the manuscript for clarity.*

The lack of a recent measurement campaign was already pointed out in lines 491-492 (surveys in the area are also quite dangerous so not very frequent). However, we included further lines in the Conclusions section (lines 494-497).

*I believe Figure 7 could be highly useful for readers attempting to understand how many scenarios fall into one regime or another. To enhance clarity, I suggest adding a reference*

*to Figure 7 in support of the discussion around line 225. This would help readers better visualize the distribution of scenarios across different regimes.*

We are sorry but in our opinion presenting Figure 7 in the position where the reviewer suggests is not useful. Figure 7 shows the domain-averaged wind strength at 10 m above the ground for each season and was originally introduced to explain the seasonal control on the probabilistic outputs of $CO_2$ concentration. In our opinion, Figure 7 does not help understanding how many scenario fall into one regime or another, an information provided clearly in lines 254-256.

*Additionally, I would like to point out that the reasoning around line 225 is based on the mean gas emission rate. Please ensure this is made explicit in the text to avoid any ambiguity.*

Done in line 229.